# Toward a realistic model of speech processing in the brain with self-supervised learning

**Juliette Millet*** [1,2,3]    **Charlotte Caucheteux*** [1,4]    **Pierre Orhan** [2]    **Yves Boubenec** [2]
**Alexandre Gramfort** [4]    **Ewan Dunbar** [2,5]    **Christophe Pallier** [6]    **Jean-Rémi King** [1,2]

*These authors contributed equally
[1]Meta AI, Paris, France    [2]Ecole Normale Supérieure, PSL University, Paris, France
[3]LPI, Université de Paris cité, Paris, France
[4]Université Paris-Saclay, Inria, CEA, Palaiseau, France
[5]University of Toronto, Toronto, Canada
[6]Cognitive Neuroimaging Unit, INSERM, Gif-sur-Yvette, France

{jumi,ccaucheteux,jeanremi}@meta.com

## Abstract

Several deep neural networks have recently been shown to generate activations similar to those of the brain in response to the same input. These algorithms, however, remain largely implausible: they require (1) extraordinarily large amounts of data, (2) unobtainable supervised labels, (3) textual rather than raw sensory input, and / or (4) implausibly large memory (e.g. thousands of contextual words). These elements highlight the need to identify algorithms that, under these limitations, would suffice to account for both behavioral and brain responses. Focusing on speech processing, we here hypothesize that self-supervised algorithms trained on the raw waveform constitute a promising candidate. Specifically, we compare a recent self-supervised model, wav2vec 2.0, to the brain activity of 412 English, French, and Mandarin individuals recorded with functional Magnetic Resonance Imaging (fMRI), while they listened to approximately one hour of audio books. First, we show that this algorithm learns brain-like representations with as little as 600 hours of unlabelled speech – a quantity comparable to what infants can be exposed to during language acquisition. Second, its functional hierarchy aligns with the cortical hierarchy of speech processing. Third, different training regimes reveal a functional specialization akin to the cortex: wav2vec 2.0 learns sound-generic, speech-specific and language-specific representations similar to those of the prefrontal and temporal cortices. Fourth, we confirm the similarity of this specialization with the behavior of 386 additional participants. These elements, resulting from the largest neuroimaging benchmark to date, show how self-supervised learning can account for a rich organization of speech processing in the brain, and thus delineate a path to identify the laws of language acquisition which shape the human brain.

## 1  Introduction

The performance of deep neural networks has taken off over the past decade. Algorithms trained on object classification, text translation, and speech recognition are starting to reach human-level performance [Xu et al., 2020]. Furthermore, the *representations* generated by these algorithms have repeatedly been shown to correlate with those of the brain [Kriegeskorte, 2015, Yamins and DiCarlo, 2016, Kietzmann et al., 2018, Kell and McDermott, 2019, Cichy and Kaiser, 2019, Toneva and

Wehbe, 2019, Millet and King, 2021, Caucheteux and King, 2022], suggesting that these algorithms converge to brain-like computations.

Such convergence, however, should not obscure the major differences that remain between these deep learning models and the brain. In particular, the above comparisons derive from models trained with (1) extraordinarily large amounts of data (40GB for GPT-2 [Radford et al., 2019], the equivalent of multiple lifetimes of reading), (2) supervised labels, which is rarely the case for humans (e.g. [Yamins and DiCarlo, 2016]), (3) data in a textual rather than a raw sensory format, and/or (4) considerable memory (e.g., language models typically have parallel access to thousands of context words to process text). These differences highlight the pressing necessity to identify architectures and learning objectives which, subject to these four constraints, would be sufficient to account for both behavior and brain responses.

Here, we hypothesize that the latest self-supervised architectures trained on raw sensory data constitute promising candidates [Borgholt et al., 2022, Bardes et al., 2021, Baevski et al., 2020]. We focus on wav2vec 2.0 [Baevski et al., 2020], an architecture that stacks convolutional and transformer layers to predict a quantization of the latent representations of speech waveforms. We train wav2vec 2.0 on 600 h of effective speech – a quantity roughly comparable to what infants are exposed to during early language acquisition (speech only makes up a small fraction of infants' daily experience) [Dupoux, 2018, Hart and Risley, 1992, Gilkerson et al., 2017].

We use standard encoding analyses [Naselaris et al., 2011, Huth et al., 2016, Yamins and DiCarlo, 2016, Kell et al., 2018] (Figure 1) to compare this model to the brains of 412 healthy volunteers (351 English speakers, 28 French speakers, and 33 Mandarin speakers) recorded with functional magnetic resonance imaging (fMRI) while they passively listened to approximately one hour of audio books in their native language [Nastase et al., 2020, Li et al., 2021] (8.5 hours of distinct audio materials in total).

To better understand the similarities between wav2vec 2.0 and the brain, we compare brain activity to each layer of this model, as well as to several variants, namely (1) a random (untrained) wav2vec 2.0 model, (2) a model trained on 600 h of non-speech sounds, (3) a model trained on 600 h of non-native speech (for example, a model trained on English speech and mapped onto the brain responses to French-speaking participants), (4) a model trained on 600 h of native speech (for example, a model trained on English speech and mapped onto the brain responses to English participants), and (5) a model trained directly on speech-to-text (i.e., a supervised learning scheme) on the native language of the participants.

Our results provide four main contributions. First, self-supervised learning leads wav2vec 2.0 to learn latent representations of the speech waveform similar to those of the human brain. Second, the functional hierarchy of its transformer layers aligns with the cortical hierarchy of speech in the brain, and reveals the whole-brain organisation of speech processing with an unprecedented clarity. Third, the auditory-, speech-, and language-specific representations learned by the model converge to those of the human brain. Fourth, behavioral comparisons to 386 supplementary participants' results on a speech sound discrimination task confirm this common language specialization.

## 2 Methods

### 2.1 Models

We train several variants of wav2vec 2.0 [Baevski et al., 2020] from scratch on different speech datasets using two different learning objectives (a self-supervised and a supervised objective).

#### 2.1.1 Architecture

Wav2vec 2.0 consists of three main modules. First, a feature encoder composed of seven blocks of temporal convolutions (output dimension 512) transforms the speech input $S$ (raw mono waveform at 16 kHz) into a latent representation $z$ (output dimension of 512, frequency 49 Hz, stride of 20 ms between each frame, receptive field of 25 ms). Second, a quantization module discretizes $z$ into $q$, a dictionary of discrete and latent representations of sounds. Third, $z$ is input to a "context network" consisting of 12 transformer blocks (model dimension 768, inner dimension 3072, and 8 attention heads), which together yield a contextualized embedding $c$, of the same dimensionality of $q$.

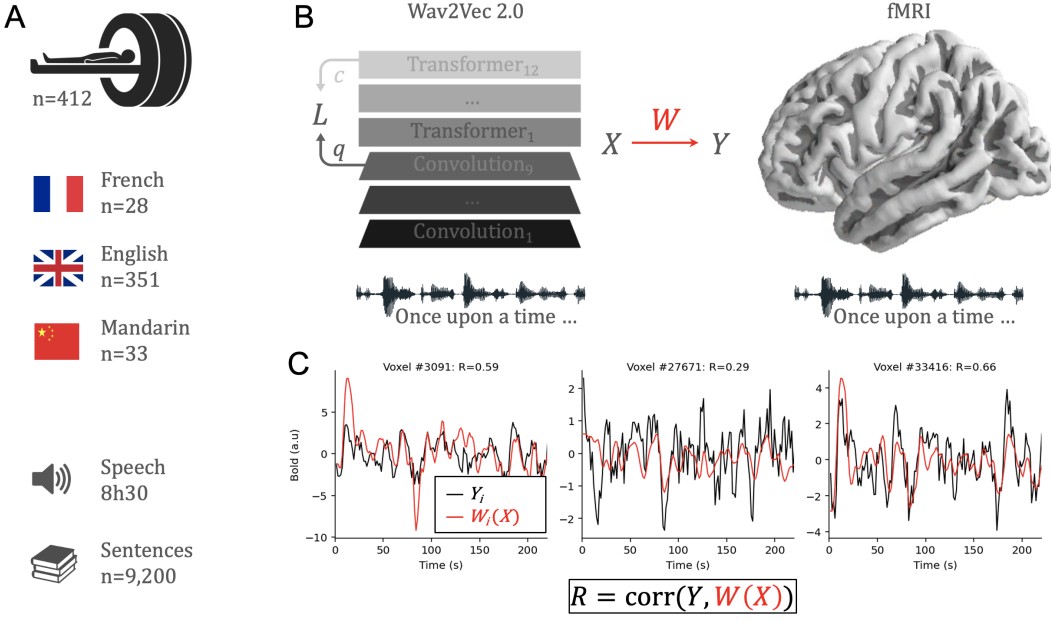

Figure 1: **Comparing speech representations in brains and deep neural networks. A.** We analyze the brain activity of 412 participants recorded with functional Magnetic Resonance Imaging (fMRI) while they passively listened to audio books in their native language (French, English or Mandarin). **B.** After training wav2vec 2.0 [Baevski et al., 2020] with self-supervised learning ($L$) over $600\,h$ of unlabelled, effective speech, we extract its activations in response to the audio books that were presented to the participants. We assess the similarity between the activations of the model $X$ and brain activity $Y$ with a standard encoding model $W$ [Nastase et al., 2020] evaluated with a cross-validated Pearson correlation $R$. **C.** Examples of the true BOLD response (black) and the predicted BOLD response (red) estimated from a linear projection of the model's activations in three voxels randomly selected from the $10^{th}$ percentile of best voxels identified by the noise ceiling analysis for the first $200\,s$ of a representative story in the test set.

### 2.1.2 Learning objective

**Self-supervised learning.** In this training paradigm, the model optimizes two losses. The first loss is contrastive and requires the model to predict the quantized representation $q$ of some masked input using $c$, from a finite set of quantized representations drawn from the input sample. The second loss ensures that the quantized representations are diverse. See Section A.2 and [Baevski et al., 2020] for details.

**Supervised learning.** In this training paradigm, the quantization module is discarded and a linear layer mapping $c$ to phonemes is added at the end of the pipeline. The model is randomly initialized and all layers (including the feature encoder) are trained using a Connectionist Temporal Classification (CTC) [Graves, 2012] loss to perform phone recognition. For both training paradigms, we extract the activations of each layer from both the feature encoder (outputting $z$) and the context network (outputting $c$). We extract the representations of the convolutional and transformer blocks using an input window of $10\,s$ of raw waveform (stride $= 5\,s$).

### 2.1.3 Training

**Datasets.** We successively train different wav2vec 2.0 models using each of four datasets: (i) the French and (ii) English CommonVoice corpora [Ardila et al., 2020], (iii) the MAGICDATA Mandarin Chinese Read Speech Corpus [Co., 2019], and (iv) a non-speech subset of the Audioset dataset [Gemmeke et al., 2017], which contains recordings of various acoustic scenes.

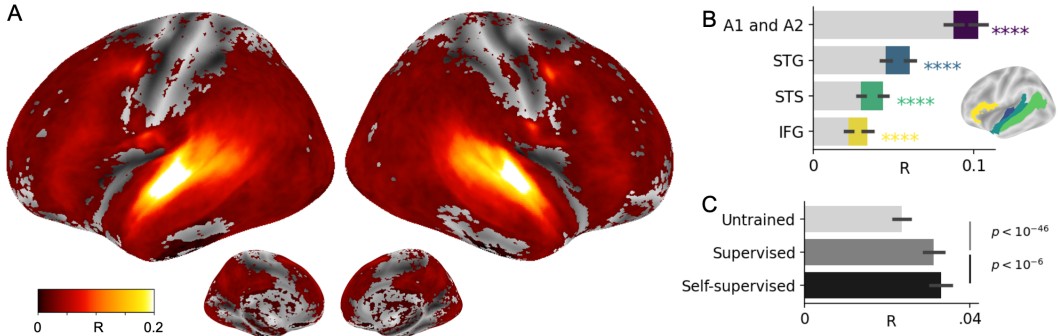

Figure 2: **Self-supervised learning suffices for wav2vec 2.0 to generate brain-like representations of speech. A.** Brain score ($R$) assessed for each subject and voxel independently, and here averaged across subjects for clarity. Only scores significantly above chance level, as assessed using a two-sided Wilcoxon test across subjects after correction for multiple comparison are color-coded ($p < 10^{-10}$). **B.** $R$ scores for the same wav2vec2 model, averaged across subjects and voxels in four brain areas typically involved during speech processing (the primary and secondary auditory cortices, the superior temporal gyrus, the superior temporal sulcus, and the infero-frontal gyrus). In grey, the brain score obtained with a randomly initialized wav2vec 2.0 architecture. Error bars are the standard errors of the mean (SEM) across subjects. The stars indicate a significant difference between the random and trained model (all $p < 10^{-4}$). **C.** $R$ scores of wav2vec 2.0 without training (top), trained with a supervised (middle) and self-supervised learning rule (bottom), on the same 600 hours of effective speech. Scores are averaged across subjects and voxels and error bars are SEM across subjects.

**Preprocessing.** All the audio datasets were randomly subsampled to have an approximate size of 600 hours, downsampled to 16 kHz and converted to mono with the Sox software[1]. We randomly split the datasets into a training (80%), a validation (10%) and a test set (10%). The audio recordings we use from the Audioset dataset are filtered so that they do not contain speech or any sounds produced by humans, such as laughter or singing. For the speech datasets, we also use their corresponding annotations (in the supervised settings). We phonemize these annotations using eSpeakNG[2]. The number of different phoneme symbols in these annotations is similar for French (32), English (39), and Mandarin Chinese (33).

**Implementation.** We train all of our models using the fairseq implementation of wav2vec 2.0[3] using default hyperparameters. We also analyze a model whose parameters were randomly initialised ("untrained" model).

We use self-supervised learning to train four models: three on the speech datasets (French, English, and Mandarin) and one on the acoustic scenes dataset. In each case, the training was performed using the same configuration file (namely, the base configuration provided in the fairseq repository for pretraining wav2vec 2.0 on LibriSpeech [Panayotov et al., 2015]). We train the models for 400k updates and select the ones with the best validation loss.

We also use the supervised training paradigm to train three models, on the French, English, and Mandarin datasets, respectively. Each training was performed using the same configuration file, which was identical to the configuration provided in the fairseq repository for fine-tuning wav2vec 2.0 on the 960 hour Voxpopuli corpus [Wang et al., 2021], except that parameters were not frozen (`freeze_finetune_updates`$= 0$) and learning was performed on all parameters of the models using the CTC loss (`feature_grad_mult`$= 0.1$). We train the models for 400k updates and we use the ones with the best word error rate (WER) on the validation set. The French model obtains 13.9 WER, the English model 28.6 WER, and the Mandarin model 4.6 WER, on their respective test sets.

---

[1]`http://sox.sourceforge.net/`
[2]`https://github.com/espeak-ng/espeak-ng`
[3]`https://github.com/pytorch/fairseq/tree/main/examples/wav2vec`

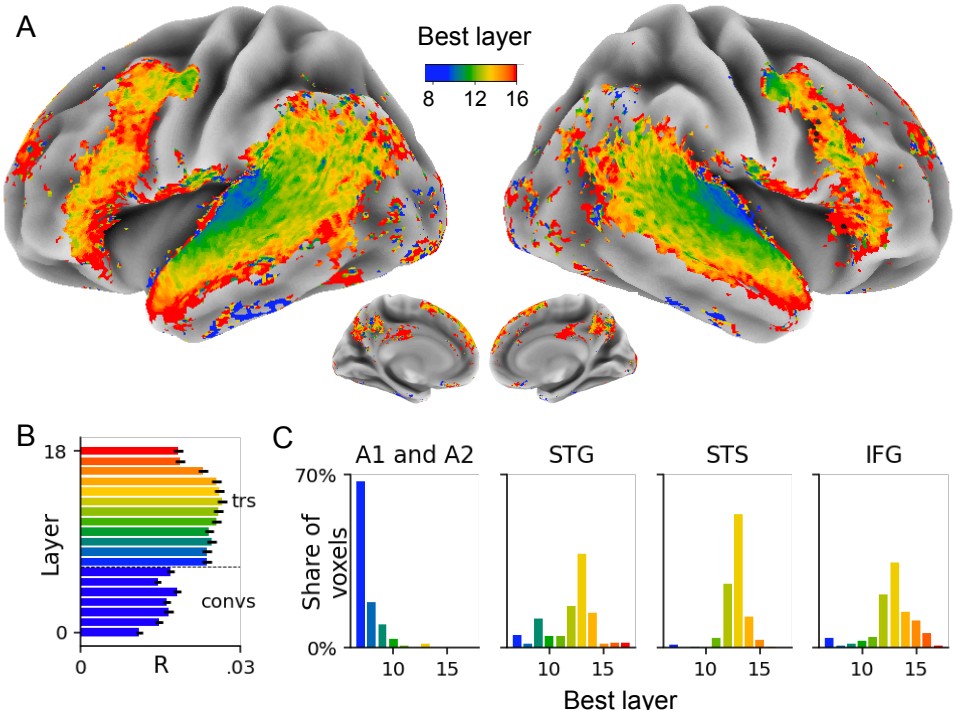

Figure 3: **The functional hierarchy of wav2vec 2.0 maps onto the speech hierarchy in the brain.** **A.** We compute the $R$ score for each layer of wav2vec 2.0 separately and estimate, for each voxel, the layer with highest brain score on average across subjects. Only the voxels with significant brain scores are displayed ($p < 10^{-18}$). While the first transformer layers (blue) map onto the low-level auditory cortices (A1 and A2), the deeper layers (orange and red) map onto brain regions associated with higher-level processes (e.g. STS and IFG). **B.** Layer-wise $R$ scores averaged across all voxels. Error bars are SEM across subjects. **C.** Proportion of voxels with most predictive layer (x-axis) in four regions typically involved in speech processing. While most voxels in the primary cortex are best predicted by the first layers of the transformer, higher-level brain areas are best predicted by deeper layers.

## 2.2 Functional MRI

We analyse a composite set of fMRI recordings aggregated from the *Little Prince* [Li et al., 2021] and the *Narratives* public datasets [Nastase et al., 2020].

**Narratives.** This dataset[4] contains the fMRI recordings of 345 native English-speaking participants listening to English narratives (4.6 hours of unique audio in total). The participants listened to different stories varying from 7 to 98 min (mean = 26 min). Following [Nastase et al., 2020], we (1) focus on fifteen representative stories and ignore the narratives that have been modified by scrambling and (2) exclude eight participants because of noisy recordings. Overall, this selection results in a curated dataset of 303 participants listening to fifteen stories ranging from 3 min to 56 min, for a total of 4 hours of unique audio (36,018 words from a vocabulary of 4,004 unique words).

**The Little Prince.** This dataset[5] contains fMRI recordings of 48 English native speakers, 33 Mandarin native speakers, and 28 French native speakers listening to *The Little Prince* in their respective native language. The experiment itself was divided into nine runs of approximately 10 min of passive listening. For each language condition, the story was read by a single native speaker. The English, Mandarin, and French audiobooks last 94, 90 and 97 minutes respectively.

---

[4]https://openneuro.org/datasets/ds002345
[5]https://openneuro.org/datasets/ds003643/versions/1.0.4

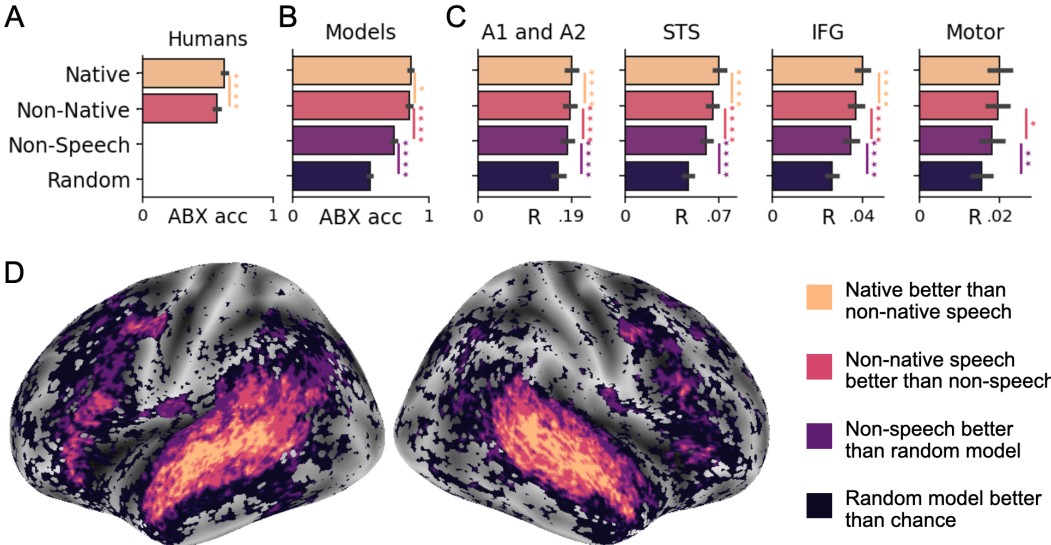

Figure 4: **The specialization of wav2vec 2.0's representations follows and clarifies the acoustic, speech, and language regions in the brain.** **A.** We first evaluate humans' language specificity by quantifying their ability to perceive phonemes of their native or non-native languages (Section 2.4) in a ABX matching-to-sample task [Schatz, 2016] (higher is better). As expected, humans are better at matching phonemes of their native language. **B.** Then, we train four wav2vec 2.0 models with self-supervised learning on four datasets – non-speech acoustic scenes, English, and French, and compute their ABX accuracy on the same speech datasets as humans. The 'random' model is wav2vec 2.0 without any training. **C.** Brain score ($R$) of each model (with an added model trained on Mandarin), averaged across voxels, in four regions of the brain (Section 2.2). **D.** Acoustic, speech and language specificity for each voxel. For instance, one voxel is considered specific to the 'native' model if its native $R$ score is higher than its 'non-native' $R$ score ($p < .05$). Only the voxels with significant $R$ scores for the untrained model are displayed ($p < 10^{-18}$). Error bars are the SEM across phone pairs in A and B, and across subjects in C. The stars indicates a significant difference between two conditions (Section 2.3).

**Preprocessing.** For Narratives, we did not perform additional preprocessing: we use the public preprocessing of the dataset already projected on the surface space ("fsaverage6") without spatial smoothing (labelled "afni-nosmooth" in the data repository). In contrast, the *Little Prince* dataset is only provided in a volumetric space. Consequently, for each language condition separately, we subselected the cortical voxels by computing a brain mask using the average of all participants' fMRI data realigned onto a common template brain via Freesurfer [Fischl, 2012]. These voxels are then projected onto a brain surface using nilearn's `vol_to_surf` function with defaults parameters [Abraham et al., 2014] and a 'fsaverage6' template surface [Fischl, 2012]. For both *Narratives* and *The Little Prince*, fMRI signals are normalized across the time dimension to have a mean of 0 and a variance of 1, for each participant, surface voxel and session independently.

**Brain parcellation.** For the purposes of certain analyses, we group the fMRI voxels into regions of interest using the Destrieux Atlas [Destrieux et al., 2010]. This parcellation results in 75 brain regions in each hemisphere. For simplicity, we label the regions as follows: A1 and A2 represents Heschl gyrus, which is the anatomical location of the primary and secondary auditory cortices, STG and STS are the superior temporal gyrus and sulcus, and IFG is the inferior frontal gyrus.

## 2.3 Brain score (R)

To quantify the similarity between the network's activations $X$ and the brain recordings $Y$, we use a standard linear encoding model [Huth et al., 2016, Yamins and DiCarlo, 2016]. For each subject, we split the data into train and test sets using a five-fold cross-validation setting. For each train split, a

linear mapping $W$ is fitted to predict the brain response $Y_{\text{train}}$ given $X_{\text{train}}$. $W$ combines a temporal alignment function with fixed weight, and a trained penalized linear regression.

**Temporal alignment.** The sampling frequency of the model's activations (between 49 and 200 Hz) differs from the sampling frequency of fMRI BOLD signals (0.5 Hz). Furthermore, the BOLD signals have delayed responses spanning over several seconds. Thus, we first convolve the model activations with a standard hemodynamic response function (HRF) using nistats [Abraham et al., 2014] `compute_regressor` function with the 'glover' model and default parameters. This results in the convolved activations $X'_{\text{train}}$ with the same sampling frequency as the fMRI $Y_{\text{train}}$ (see A.3).

**Penalised linear regression.** Once temporally aligned, we fit an $\ell_2$-penalised linear regression that predicts the brain signals $Y_{\text{train}}$ given the activations $X_{\text{train}}$. We use the `RidgeCV` function from scikit-learn [Pedregosa et al., 2011], with the penalization hyperparameter $\lambda$ varying between 10 and $10^8$ (20 values scaled logarithmically) chosen independently for each dimension with a nested cross-validation over the training set (see A.4).

**Evaluation.** We evaluate the linear mapping $W$ on the held out sets by measuring Pearson's correlation between predicted and actual brain responses: $R = \text{corr}(Y_{\text{test}}, W \cdot X_{\text{test}})$. Finally, we average the correlation scores across test splits to obtain the final "brain score". To report the average layer $k^*$ with the highest brain score for each voxel (Figure 3), while being robust to regression-to-the-mean biases, we first find the best layer $k_s$ for each participant $s$ and each voxel independently and then compute a circular mean across the $N = 412$ participants and the $K = 19$ layers:

$k^* = \text{angle}\left(\frac{1}{N}\sum_{s=1}^{N}\exp\left(\frac{2i\pi k_s}{K+1}\right)\right)$

**Statistics.** We assess the reliability of brain scores with second-level analyses across participants thanks to a Wilcoxon signed-rank test across participants. Thus, the resulting p-values are not affected by fMRI auto-correlation within participants. We perform statistical correction for multiple comparisons with Benjamini–Hochberg False Discovery Rate (FDR) across voxels [Benjamini, 2010].

## 2.4 Behavioral experiment

To compare the phonetic representations of our models to those of humans, we compare the forced-choice discrimination judgements of online participants[6] to an analogous method applied to wav2vec 2.0 [Schatz, 2016]. Specifically, for each triplet of sound "ABX", participants judged whether the stimulus X was more similar to A or B. Analogously, we computed the Euclidean distance in the most discriminative layer of wav2vec 2.0 (here transformer layer 5) to determine whether X was closer to A or B. Additional data, analyses and model-human comparison can be found in [Millet and Dunbar, 2022]. We focus on the French and English stimuli, which represent $\approx$ 6,000 ABX triplets (testing 508 English and 524 French phone pairs), with 386 participants in total (193 from each language group).

In Figure 4-A, we report the ABX accuracy of English- and French-speaking participants in both their native and non-native language (either English or French). We first average results per phone pair, and then average over phone pairs to obtain the ABX discrimination accuracy. Similarly, in Figure 4-B, we compute the ABX accuracy of our wav2vec 2.0 models on the same evaluation sets as the participants, using the parameters described in [Millet and Dunbar, 2022]. English and French models are evaluated on the same ('native') or different ('non-native') language stimuli as their training. The random and non-speech models are evaluated on both French and English speech stimuli.

## 3 Results

**Wav2vec 2.0 maps onto brain responses to speech.** We estimate whether the activations of wav2vec 2.0 models linearly map onto the human brain activity of 412 individuals listening to audio books in the fMRI scanner. For this, we first independently train three models with 600 h of

---

[6]https://docs.cognitive-ml.fr/perceptimatic/

French, English, or Mandarin, respectively, and compute the brain scores ($R$) with the corresponding participants. Specifically, we (1) convolve the activations ($X$) of the model with a hemodynamic response function (HRF), (2) train a $\ell_2$-penalized linear regression on a training split to map them to brain activity $Y$, and (3) compute the Pearson correlation coefficient between (i) the true fMRI activity and (ii) the predicted activations on a test split. The models' activations significantly predict brain activity in nearly all cortical areas, reaching the highest $R$ scores in the primary and secondary auditory cortices (Figure 2-A B). These scores are significantly higher than those obtained with a randomly initialised model ($p < 10^{-50}$ on average across voxels), and this comparison is robust across language groups (all $p < 10^{-5}$).

**Comparison of self-supervised to supervised models.** Does self-supervision reach representations that are as brain-like as those obtained with supervised learning? To address this issue, we trained wav2vec 2.0 with an alternative, supervised objective, namely, predicting phonetic annotations from the same 600 hours of effective speech sounds. We then implemented the $R$ score analyses described above. The results show that self-supervised learning in fact leads to modestly but significantly better $R$ scores than supervised learning (Figure 2-C): $\Delta R = 0.002, p < 10^6$.

**The hierarchy of wav2vec 2.0 maps onto the hierarchy of the cortex.** To compare the speech hierarchy in the brain with the functional hierarchy learned by wav2vec 2.0, we evaluate the $R$ score of each layer of the model (Figure 3). First, we observe that convolutional layers are less predictive than transformer layers. Second, within the transformers, the hierarchy of representations aligns with the expected cortical hierarchy [Hickok and Poeppel, 2007]: while low-level areas (A1, A2) are best predicted by the first transformer layers, higher level areas (IFG, STS) are best predicted by deeper layers. Remarkably, this hierarchy extends to supplementary motor and motor areas in both hemispheres (Figure 3-A).

**Language specificity in phone discrimination tasks.** The acoustic features underlying speech (fricatives, vowels, and so on) may also characterize non-speech sounds (the sound of tree leaves in the wind, of a stone falling, and so on). Does the model show commonalities merely with general auditory processing in the brain, or does it capture speech-specific processing? If so, does it show commonalities with brain representations that are specific to the native language of the participants, or merely to general speech processing? We first evaluate the specialization of humans' perception to their native language using an ABX behavioral task (Section 2.4). Specifically, we compare 386 French and English participants on their ability to distinguish native and non-native phones. As expected [Bohn, 2017, Kuhl et al., 2005], participants were better at discriminating native sounds than non-native ones (across phone pairs: $p < 10^{-18}$, Figure 4-A). Second, applying the same test to our self-supervised French and English models shows that, like humans, models best discriminate sounds from their 'native' language (i.e., the French model better distinguishes French stimuli than English ones, across phone pairs, and vice versa: $p < 0.05$). Interestingly, the ABX accuracy of the model is significantly higher than participants'. This quantitative difference may be partially explained by the fact that participants – and online participants in particular – undergo fluctuating attention, and adopt strategies which can negatively impact performance [Humphreys, 1939]. Finally, as expected, the random and acoustic models obtain the worst ABX accuracy. Overall, These results confirm that 600 h of self-supervised learning on effective speech suffices for wav2vec 2.0 to learn language-specific representations (Figure 4-B).

**Wav2vec 2.0 and the brain learn language specific representations.** Next, we compare the brain scores of random, non-speech, non-native and native models (Figure 4-C D). First, our results show that the non-speech model attains higher $R$ scores than the random model (on average across voxels, $\Delta R = 0.006, p = 10^{-31}$) confirming the importance of learning to generate brain-like representations. Second, non-native models attain higher $R$ scores than the non-speech model ($\Delta R = 0.002, p = 10^{-9}$), confirming that wav2vec 2.0 learns speech-specific representations of sounds when trained on speech. Finally, the native model attains higher $R$ scores than non-native models ($\Delta R = 0.002, p = 10^{-15}$).

# 4 Discussion

Human infants acquire language with little to no supervision: A few hundred hours of speech suffices for their young brain to learn to discretize phonemes, segment morphemes, and assemble words in the language(s) of their social group [Dupoux, 2018, Gilkerson et al., 2017]. However, the learning principle that allows this unique feat remains, to date, unknown.

Here, we test whether self-supervised learning applied to a limited amount of speech effectively accounts for the organization of speech processing in the human brain as measured with fMRI. For this, we train several variants of wav2vec 2.0 [Baevski et al., 2020] with three curated datasets of French, English, and Mandarin, and compare their activations to those of a large group of French, English, and Mandarin speakers recorded with fMRI while passively listening to audio stories. Our results show that this self-supervised model learns (i) representations that linearly map onto a remarkably distributed set of cortical regions (Figure 2), (ii) a computational hierarchy that aligns with the cortical hierarchy (Figure 3), and (iii) features specific to the language of the participants (Figure 4).

**Towards a biologically-plausible learning principle.** These results extend recent findings on the similarities between the brain and a variety of deep learning models trained with biologically-implausible objectives and data. First, fMRI [Kell et al., 2018, Millet and King, 2021, Thompson et al., 2021], electroencephalography [Huang et al., 2018], and multi- or single-unit responses to sounds [Koumura et al., 2019, Begus et al., 2022] have been shown to be linearly predicted by the activations of deep convolutional networks trained on *supervised* auditory tasks. For example, [Millet and King, 2021] showed that a supervised speech-to-text model better accounted for brain responses to speech in 102 individuals when it was trained on speech recognition rather than auditory scene classification. Similarly, [Kell et al., 2018] showed that eight participants listening to brief speech and non-speech sounds demonstrated fMRI responses in the temporal lobe that aligned with those of a deep convolutional neural network trained on a binary auditory classification task. Our results, based on up to 50 times more fMRI recordings of the entire cortex show that such representational similarities hold with a self-supervised objective [Lerner et al., 2011, Berezutskaya et al., 2017, Caucheteux et al., 2021c,b]. Second, a growing series of MEG [Toneva and Wehbe, 2019, Caucheteux and King, 2022], fMRI [Mitchell et al., 2008, Qian et al., 2016, Pereira et al., 2018, Schwartz et al., 2019, Antonello et al., 2021, Jain and Huth, 2018] and electro-physiology studies [Schrimpf et al., 2021, Goldstein et al., 2022] showed that text-based language models trained on very large corpora generate brain-like representations too. While these results suggest elements of convergence between language models and the brain [Caucheteux and King, 2022], they also remain biologically implausible: not only are these algorithms pre-equipped with abstract linguistic units such as characters and words, but they are trained on corpora that no one would ever be able to read in their lifetime. In contrast, wav2vec 2.0 is here trained with a reasonable amount of raw speech waveforms [Hart and Risley, 1992, Gilkerson et al., 2017, Dupoux, 2018]. The functional similarity between wav2vec 2.0 and the brain thus opens the way to clarify how humans learn to process speech.

**The emergence of a brain-like hierarchy of speech processing.** The present study reveals the hierarchical organization of speech processing with remarkable clarity. First, the functional hierarchy learnt by wav2vec 2.0 is aligned with the anatomy: *e.g.* the superior temporal sulcus and the temporal pole are known to project to the ventral and dorsal part of the inferofrontal gyrus, respectively [Petkov et al., 2015]. Second, the identification of functional gradients within the prefrontal cortex, and down to the motor areas typically associated with larynx and mouth control [Dichter et al., 2018] reinforces the relevance of motor processes to speech perception [Kellis et al., 2010, Mugler et al., 2014, Shamma et al., 2021]. Finally, the existence of multiple levels of representations around the inferofrontal cortex is consistent with the idea that Broca's area may be responsible for merging linguistic units [Chomsky, 2000, Friederici, 1999, Hagoort, 2005, Poeppel et al., 2012]. It should be noted, however, that our results aggregate a large cohort of individuals which could mask a more modular organization at the individual level.

**Interpreting the neural representations of speech.** Interpreting neural representations is a notoriously difficult challenge to both AI and neuroscience. Here, we first investigate language specificity and show that the neural representations specific to the native models are primarily represented

in the superior temporal sulcus and middle temporal gyrus (Figure 4D): areas known to represent phonetic features [Mesgarani et al., 2014]. However, these effect are relatively modest (Figure 4): the random model and the non-speech model reach, in STS and STG, 67% and 87% of the brain scores obtained by the "native" model, respectively. While this high baseline initially surprised us, this phenomenon could be explained by the fact that the auditory cortex is continuously bombarded by – and should thus be tuned to – non-speech input. Second, our probing analyses show that the models trained with self-supervised learning learn relevant acoustic and linguistic representations (Supplementary Figure S1). This result, consistent with Vaidya et al. [2022] and Stephenson et al. [2019], suggests that the difference of brain scores observed between the random, non-native and native models (Figure 4) may be partly driven by the corresponding spectro-temporal, phonetic, word and sentence-level representations, respectively. These elements of interpretation remain, however, scarce, and a systematic interpretation of the representations shared between wav2vec 2.0 and the brain remains necessary.

**Scope of the study.** It is important to stress that the scope of the present study could be broadened in several ways. First, our study focuses on adult speakers, whose cultural and educational background is not representative of the population [Henrich et al., 2010]. Second, we focus on the passive listening of three languages. Third, we focus on one self-supervised learning architecture [Baevski et al., 2020], and its functional alignment with fMRI, whose temporal resolution is notoriously limited. Generalizing the present approach to more languages [Malik-Moraleda et al., 2022], a larger spectrum of children and adult participants recorded with a variety of electrophysiological and neuroimaging devices will thus be essential to confirm, precise, and/or mitigate the present findings.

**The remaining gap between brain and speech models.** Several major gaps can be evidenced between wav2vec 2.0 and the brain. First, the transformer layers are not temporally constrained: each layer can access all elements within the contextual window. This differs from the necessarily recurrent nature of processing in the brain. Second, wav2vec 2.0 behaves differently to humans in specific tasks. In particular, it is overly-sensitive to band-pass filtering, non-robustly exploit fine temporal structures [Weerts et al., 2021] and fails to display the expected categorical responses [Millet et al., 2021]. Third, recent studies show that wav2vec 2.0 encodes significantly less semantic information than text-based models [Pasad et al., 2021, Vaidya et al., 2022]. While our analyses suggest that learning allows wav2vec 2.0 to capture some lexical features in its deep layers (Figure S1, Table S4), it remains unclear whether these layers also capture complex syntactic structures, such as recursive syntactic trees [Lakretz et al., 2021, Caucheteux et al., 2021a]. We speculate that these limitations may be due to the time scales of wav2vec 2.0 which, unlike humans, learns very short-term representations of speech. In any case, these differences likely explain why the brain scores of wav2vec 2.0 remain substantially lower than our noise-ceiling (19% on average, and up to 74% in Heschl's gyrus and sulcus, Table S1, Figure S2).

Overall, the complexity of the human brain is often thought to be incompatible with a simple theory: "Even if there were enough data available about the contents of each brain area, there probably would not be a ready set of equations to describe them, their relationships, and the ways they change over time" [Gallant, 2013]. By showing how the equations of self-supervised learning give rise to brain-like processes, this work contributes to challenge this view.

## Acknowledgments

This project was funded, in part, by the Bettencourt-Schueller Foundation, the Philippe Foundation, and FrontCog grant ANR-17-EURE-0017 to JRK for his work at PSL.

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
