# A  Appendix

## A.1  Self-supervised loss formula

Wav2vec 2.0, when trained in a self-supervised way, uses a loss ($L$) which is the weighted combination of two losses: one diversity loss ($L_d$), which pushes the quantization module to contain representations that are as diverse as possible, and one Contrastive Predictive Coding loss ($L_m$), which pushes the model to choose, from the context network output $c$, the right quantized representation ($q$) of some masked input, among other possible representations. $L_m$ has the following formula, for some masked time step $t$:

$$\mathcal{L}_m = -\log \frac{\exp\left(\mathrm{sim}\left(\mathbf{c}_t, \mathbf{q}_t\right)/\kappa\right)}{\sum_{\tilde{\mathbf{q}} \sim \mathbf{Q}_t} \exp\left(\mathrm{sim}\left(\mathbf{c}_t, \tilde{\mathbf{q}}\right)/\kappa\right)} \tag{1}$$

with $\mathrm{sim}(\mathbf{a}, \mathbf{b}) = \mathbf{a}^T\mathbf{b}/\|\mathbf{a}\|\|\mathbf{b}\|$, $\kappa$ the temperature, which is constant during training, $Q_t$ the set of $K + 1$ quantized candidate the model has to choose from, including the right one, i.e. $q_t$.

$L_d$ is included to encourage the equal use of the $V$ possible entries of each of the $G$ codebooks of the quantization module. The goal is to maximize the entropy of the averaged softmax distribution over the codebook entries for each codebook $\bar{p}_g$, across a set of utterances:

$$\mathcal{L}_d = \frac{1}{GV} \sum_{g=1}^{G} -H\left(\bar{p}_g\right) = \frac{1}{GV} \sum_{g=1}^{G} \sum_{v=1}^{V} \bar{p}_{g,v} \log \bar{p}_{g,v} \tag{2}$$

## A.2  Supervised loss formula

When trained in a supervised way, wav2vec 2.0 is trained to optimise a Connectionist Temporal Classification loss parameterized over $\theta$:

$$\mathrm{argmin}_\theta \ -\log \sum_{a \in a_{U,V}} \prod_{t=1}^{d_t} p_{\mathrm{CTC}}\left(a_t \mid m_\theta(U)\right) \ , \tag{3}$$

where $m_\theta(U) \in \mathbb{R}^{d_\tau \times d_v}$ are the probabilistic predictions of the model at each $\tau$ time sample given the input raw waveform $U \in \mathbb{R}^{d_\tau \times d_u}$, $V \in \mathbb{R}^{d_t \times d_v}$ are the true transcriptions of $U$, and $a_{U,V}$ is the set of all possible alignments between $U$ and $V$.

## A.3  Preprocessing of the model's activations

The activations of the network $X \in \mathbb{R}^{d_{\hat{i}} \times d_x}$ are first normalized to be between $[0, 1]$ for each listening session. Then, we use nistats [Abraham et al., 2014] `compute_regressor` function with the 'glover' model to temporally convolve ($h \in \mathbb{R}^{d_{\hat{i}}}$) and temporally down-sample ( using $g : \mathbb{R}^{d_{\hat{i}}} \to \mathbb{R}^{d_t}$) each artificial neuron $j$:

$$\hat{x}^{(j)} = g\left(x^{(j)} * h\right) \ . \tag{4}$$

## A.4  Penalized linear model - Ridge regression

For each split $s$, we fit an $\ell_2$-penalized linear model $V \in \mathbb{R}^{d_x \times d_z}$ trained to predict the transformed BOLD time series from the model activations for each dimension independently. The formula of the optimization is the following:

$$\mathrm{argmin}_V \sum_{i \in \mathrm{train}_s} (V^\top \hat{X}_i - y_i)^2 + \lambda\|V\|^2 \ . \tag{5}$$

## A.5  Probing the linguistic features encoded in wav2vec2 activations

Interpreting the representations of deep learning models is notoriously difficult. To address this issue, [Pasad et al., 2021] explored the encoding of local acoustic features, phone identity, word identity and word meaning across layers. Similarly, [Millet et al., 2021] compared representations

to human behavioural data to assess whether they better captured listeners' perception of higher-level phonemic properties or of lower-level subphonemic properties of speech stimuli. Finally, [Vaidya et al., 2022] recent study explores filter banks, spectrograms, phonemes and words across layers. Here, we complement these analyses by showing that self-supervised learning allows wav2vec 2.0 to learn represents, along its hierarchy the representations of MEL spectrograms, phonetic categories and word embeddings (Figure S1).

For this, we perform a ridge regression on the Timit dataset[7] to predict five auditory and linguistic features from the activation functions of each layer and model of the present paper. We study the following features:

- the MEL spectrogram of the audio, computed using librosa (d=128)

- the phonemes (categorical features). We use the transcripts and alignments provided in Timit.

- the word embedding and part-of-speech of the words. The time alignments for words are provided by Timit. We use spaCy to compute the word embedding (medium model, d=300), and their part-of-speech (categorical feature, d=19).

- the sentence embedding of each sample, provided by Laser.

We use a subset of 1,680 samples from Timit, each sample being an audio recording of a short sentence (<10 seconds) from 24 speakers. The model's activations were mean-pooled to the sampling rate of each feature.

The results show that the layers of wav2vec 2.0 partially follow the hierarchy predicted from neuro-linguistics [Hickok and Poeppel, 2007] (Table S4): the first layers of the transformer best account for the spectro-temporal information, whereas deeper layers best account for the phonetic, word-level and sentence level information. While all of these features emerge with training (Figure S1), only the highest-level features (phone, word and sentence-level) appear to be specific to speech and to the language with which wav2vec 2.0 was trained (Figure S1).

Interestingly, the word and sentence-level features are encoded deeper in the supervised network (best layer=18 in Table S4) compared to the unsupervised network (best layer=14), which suggests that self-supervised learning generates a reservoir representations in its middle layers, reservoir which may partly overlap with the labels used in supervised learning. Together with our ABX tests, and layer-wise tuning of each voxel (Figure 3), these elements suggest that the representations of speech shaped by our experience are learnt and instantiated in the superior temporal gyrus and sulcus. These elements, consistent with previous electrophysiological studies [Mesgarani et al., 2014], thus provide a coherent spectrum of evidence for the location of acquired speech representations in the brain.

### A.6 Noise ceiling analysis

The noise in fMRI recordings is inevitable. To estimate the maximum explainable signal given this level of noise, we follow previous studies and employ a shared-response model, or "noise ceiling" [Huth et al., 2016, Caucheteux and King, 2022, Caucheteux et al., 2022]. Precisely, we predict the brain signals of one subject given the brain activity of the other subjects, in response to the same audio recording. In practice, we apply the same evaluation as Equation (2.3), for one subject $s$ and one voxel $v$, but we use the average brain signals of other subjects' brains $\overline{Y}^{(s)} = \frac{1}{|S|} \sum_{s' \neq s} Y^{(s')}$ instead of the activations $X$. As a result, the "noise ceiling" of one subject ($s$) and one voxel ($v$) is computed as follows:

$$R_{\text{noiseceil}} = \text{Corr}\big(W \cdot \overline{Y}^{(s)}, Y^{(s,v)}\big) \quad , \tag{6}$$

where $W$ is an $\ell_2$-penalized linear regression fitted on separate train data, using a cross validation setting with five test folds.

We compute such noise ceiling on 290 subjects of the Narrative dataset listening to the same stories (Figure S2). We report the noise ceiling across voxels in Figure S2, and, in Table S1, the brain scores of the networks studied in the main paper normalised by the noise ceiling. Precisely, for each

---

[7]https://catalog.ldc.upenn.edu/LDC93S1

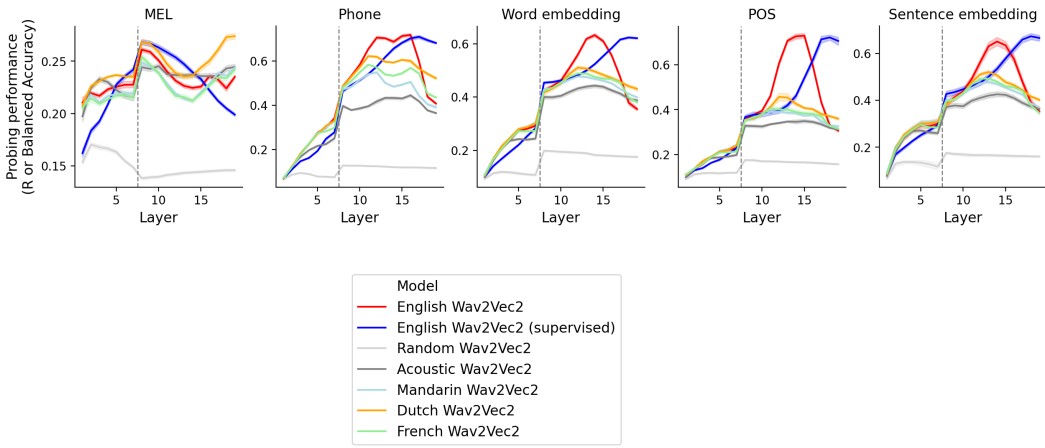

Figure S1: **Linguistic features encoded in each layer of the networks.** For each layer of each network, we train a l2-penalized linear model from scikit-learn [Pedregosa et al., 2011] to predict several linguistic categories given the embedding. The tested categories are the following: MEL (the MEL spectrogram of the audio, d=128), phone (the phoneme, categorical, d=39), the word embedding of the word (computed with spaCy (`https://spacy.io`) English model, d=300), the Part-Of-Speech (POS) of the word provided by spaCy (categorical feature, n=19), and the embedding of the sentence, computed using Laser (`https://github.com/facebookresearch/LASER`) (d=1,024). We train and test the linear probe on a subset of Timit data (`https://catalog.ldc.upenn.edu/LDC93s1`), using a 10-folds cross-validation scheme, and report the probing accuracy (either $R$ for continuous variables or balanced accuracy for categorical variables) for each possible target feature. We average the corresponding probing performances across the 10 folds. Error bars are standard errors of the mean across folds.

voxel, we divide the average brain scores by the noise ceiling for this particular voxel. While low on average, the unsupervised wav2vec2 model reaches 74% of the noise ceiling in Heschel, and more than 20% in STS, STS and IFG.

|                      | Average | Top10  | Heschl | STG    | STS    | IFG    | Motor  |
|----------------------|---------|--------|--------|--------|--------|--------|--------|
| Random wav2vec2      | 13.9%   | 29.0%  | 66.9%  | 32.0%  | 21.8%  | 15.9%  | 11.9%  |
| Non-Speech           | 16.4%   | 33.9%  | 71.0%  | 36.8%  | 26.9%  | 19.0%  | 11.7%  |
| Non-Native           | 17.6%   | 35.9%  | 73.0%  | 39.0%  | 29.1%  | 21.0%  | 12.9%  |
| Native, Supervised   | 18.3%   | 36.7%  | 74.2%  | 39.6%  | 29.8%  | 21.2%  | 13.6%  |
| Native, Unsupervised | 18.8%   | 37.9%  | 74.4%  | 40.3%  | 31.3%  | 22.8%  | 13.8%  |
| Noise ceiling        | 100.0%  | 100.0% | 100.0% | 100.0% | 100.0% | 100.0% | 100.0% |

Table S1: **Brain scores *with* noise ceiling normalisation**. Brain scores divided by the noise ceiling, for the Narrative dataset, on average across all voxels ('Average'), for the 10% best voxels of the noise ceiling ('Top10', Figure A.6) and the voxels of five regions of interests.

Below, we report the brain scores of our models, normalised by such noise ceiling. Precisely, we compute the brain scores for each subject and voxels

|  | Average | Top10 | Heschl | STG | STS | IFG | Motor |
|---|---|---|---|---|---|---|---|
| Random wav2vec2 | 0.019 | 0.069 | 0.192 | 0.071 | 0.044 | 0.024 | 0.011 |
| Non-Speech | 0.022 | 0.080 | 0.205 | 0.081 | 0.055 | 0.028 | 0.011 |
| Non-Native | 0.024 | 0.085 | 0.211 | 0.086 | 0.059 | 0.031 | 0.012 |
| Native, Supervised | 0.025 | 0.086 | 0.213 | 0.087 | 0.060 | 0.032 | 0.013 |
| Native, Unsupervised | 0.025 | 0.089 | 0.214 | 0.089 | 0.063 | 0.034 | 0.013 |
| Noise ceiling | 0.117 | 0.219 | 0.287 | 0.181 | 0.196 | 0.149 | 0.094 |

Table S2: **Brain scores *without* noise ceiling normalisation** Same as Table S1, but without dividing by the noise ceiling estimates.

|  | Avg | Top10NoiseCeil | Heschl | STG | STS | IFG | Motor |
|---|---|---|---|---|---|---|---|
| Unsupervised | 0.03 +/- 0.001 | 0.09 +/- 0.002 | 0.21 +/- 0.007 | 0.09 +/- 0.003 | 0.06 +/- 0.002 | 0.03 +/- 0.001 | 0.01 +/- 0.001 |
| Supervised | 0.02 +/- 0.001 | 0.09 +/- 0.002 | 0.21 +/- 0.007 | 0.09 +/- 0.003 | 0.06 +/- 0.002 | 0.03 +/- 0.001 | 0.01 +/- 0.001 |
| Noise ceiling | 0.12 +/- 0.006 | 0.22 +/- 0.006 | 0.29 +/- 0.008 | 0.18 +/- 0.006 | 0.20 +/- 0.006 | 0.15 +/- 0.006 | 0.09 +/- 0.006 |
| Ratio | 0.19 +/- 0.006 | 0.38 +/- 0.010 | 0.74 +/- 0.025 | 0.40 +/- 0.013 | 0.31 +/- 0.011 | 0.23 +/- 0.010 | 0.14 +/- 0.014 |

Table S3: Brain scores and noise ceiling estimates. Ratio indicate the unsupervised model divided by the noise ceiling. Scores are averaged across subjects and either all the voxels ('Avg') or the voxels of the selected regions of interests.

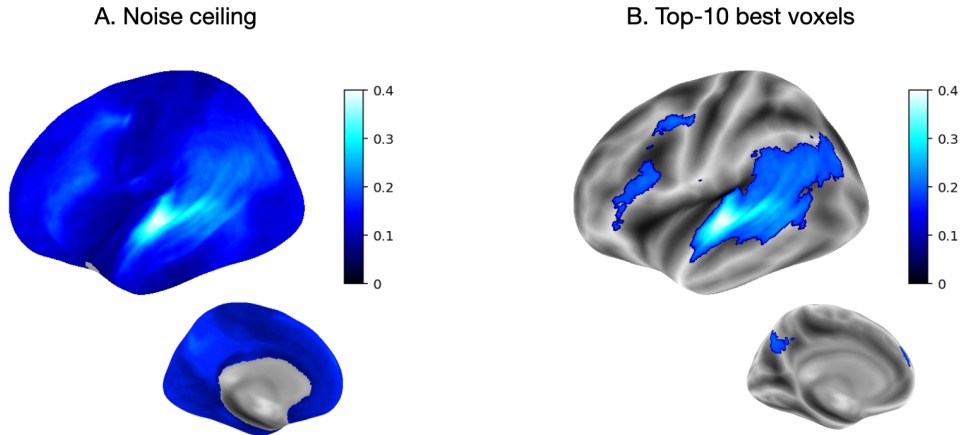

Figure S2: **Noise ceiling. A.** Noise ceiling estimates computed on 290 subjects of the Narratives dataset, averaged across subject. We only display the significant voxels across subjects ($p < 10^{-18}$). **B.** Same as A, but we only display the 10% voxels with the best noise ceiling estimates on average across subjects.

|  | MEL | Phone | Wordemb | POS | Sentemb | Average |
|---|---|---|---|---|---|---|
| Random wav2vec2 | 2.0 | 8.7 | 8.0 | 8.9 | 8.1 | 7.1 |
| Acoustic wav2vec2 | 12.5 | 15.7 | 14.0 | 14.4 | 14.2 | 14.2 |
| Mandarin wav2vec2 | 9.1 | 11.9 | 12.2 | 11.9 | 13.0 | 11.6 |
| French wav2vec2 | 8.0 | 11.0 | 12.7 | 11.8 | 13.0 | 11.3 |
| Dutch wav2vec2 | 18.9 | 11.4 | 12.0 | 12.4 | 13.0 | 13.5 |
| English wav2vec2 | 8.0 | 15.2 | 14.0 | 14.4 | 14.0 | 13.1 |
| English wav2vec2 (supervised) | 8.0 | 16.9 | 18.0 | 18.0 | 18.0 | 15.8 |
| Avg | 9.5 | 13.0 | 13.0 | 13.1 | 13.3 | 12.4 |

Table S4: For each model (row) and target (column), the layer that maximizes probing performance (Figure S1), averaged across the 10 cross-validation folds.

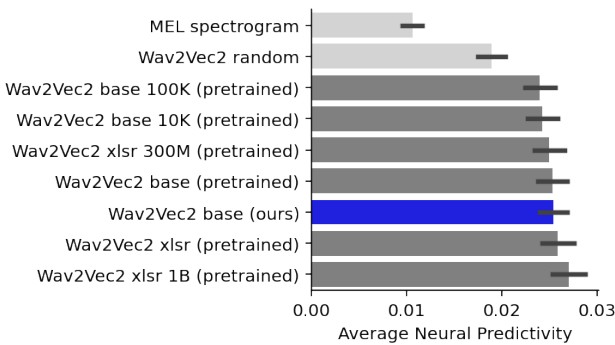

Figure S3: **Brain scores of self-supervised pre-trained models.** Brain scores, averaged across all voxels and subjects, for the MEL spectrogram, a wav2vec2 (base) architecture with random weights, wav2vec 2.0 (base) pre-trained with self supervised learning on 100K hours from Voxpopuli (Wang, 2021) ('wav2vec2-base-100k-voxpopuli' from huggingface), on 10K hours from Voxpopuli ('wav2vec2-base-10k-voxpopuli'), on 53K hours of english ('wav2vec2-base'), two models pre-trained on the same multilingual corpus of 436K hours, with 300M ('wav2vec2-xls-r-300m') and 1B parameters ('wav2vec2-xls-r-1b'), respectively, and our model trained on 600 hours of english speech (in blue). +/- refers to standard errors of the mean across subjects.

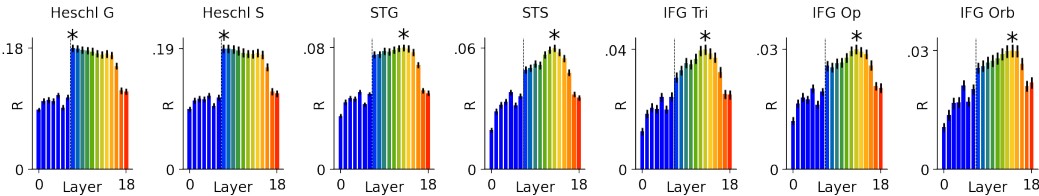

Figure S4: **Brain scores for each layer of wav2vec 2.0** Same as figure 3B, but for different regions of the brain. Brain scores are averaged across all voxels in each regions.