# OpenReview forum: "Toward a realistic model of speech processing in the brain with self-supervised learning"
_NeurIPS.cc/2022/Conference — NeurIPS 2022 Accept_

### Official Review · Reviewer_HY2z · 2022-06-28

**Rating:** 7
**Confidence:** 2
**Soundness:** 3 good
**Presentation:** 3 good
**Contribution:** 3 good

**Summary:**

The paper provides a study to investigate how Wav2Vec 2.0, as a self-supervised method, learns brain like representation. Different sets of experiments were conducted to examine the cortical hierarchy of speech processing aligns its function. It also provides observation on brain's reaction to non-speech, non-native and native sounds compared to that of Wav2vec 2.0.

**Questions:**

- I believe the results are interesting to be shared, but I do not see any machine learning innovation in this paper. Basically the authors have used an already proposed self-supervised method to compare with the brain action. I believe this could be a better fit for some speech related journals.
- R is some times used as Neural predictivity score and sometimes as Pearson correlation coefficient. Please make it clear through the paper.
- R is some times very small, e.g. Figure 4 B. is small R still something to rely on and make conclusion for?
- There are some related work, e.g. https://www.science.org/doi/abs/10.1126/science.1245994, that could be mentioned in the paper.
- Although the results are promising, please make the conclusion linked to the current set up and do not strongly generalize it to all cases.

**Limitations:**

no limitation

**Strengths And Weaknesses:**

Strengths:
Interesting observations and an extensive set of results
Weakness:
No machine learning innovation

---

> ### Author Response · Authors · 2022-08-02
> **Main Answer**
>
> We thank Reviewer 3 for their thorough review, as well as for their valuable remarks, which we will address below.
>
> > “I believe the results are interesting to be shared, but I do not see any machine learning innovation in this paper. Basically the authors have used an already proposed self-supervised method to compare with the brain action. I believe this could be a better fit for some speech related journals.”
>
> As specified on the [Neurips guidelines](https://nips.cc/Conferences/2022/CallForPapers), the contributions of a submission need not be a new machine learning technique, but can target neuroscience (and “neural coding” in particular) as well as “machine learning for sciences”.
>
> We believe that this study effectively provides such a contribution: it shows that self supervised learning can provide an effective framework to quantifiably account for the functional organization of speech processing in the human brain. This finding is important, as speech is a trait uniquely developed in the human species, and this cognitive faculty, directly dependent on our cultural environment, is learnt largely without supervision.
>
> > “R is some times used as Neural predictivity score and sometimes as Pearson correlation coefficient. Please make it clear through the paper.”
>
> We thank you for noticing this lack of homogeneity. We will now correct the manuscript to use a single terminology.
>
> > “R is some times very small, e.g. Figure 4 B. is small R still something to rely on and make conclusion for?”
>
> This is a good point. We now computed the noise ceiling on 290 subjects who listened to the very same audio stories, and can thus provide a basis for a simple “noise ceiling” analysis.
> |               | Avg            | Top10   | Heschl         | STG            | STS            | IFG            | Motor          |
> |:--------------|:---------------|:-----------------|:---------------|:---------------|:---------------|:---------------|:---------------|
> | Unsupervised  | 0.03 +/- 0.001 | 0.09 +/- 0.002   | 0.21 +/- 0.007 | 0.09 +/- 0.003 | 0.06 +/- 0.002 | 0.03 +/- 0.001 | 0.01 +/- 0.001 |
> | Supervised    | 0.02 +/- 0.001 | 0.09 +/- 0.002   | 0.21 +/- 0.007 | 0.09 +/- 0.003 | 0.06 +/- 0.002 | 0.03 +/- 0.001 | 0.01 +/- 0.001 |
> | Noise ceiling | 0.12 +/- 0.006 | 0.22 +/- 0.006   | 0.29 +/- 0.008 | 0.18 +/- 0.006 | 0.20 +/- 0.006 | 0.15 +/- 0.006 | 0.09 +/- 0.006 |
> | Ratio         | 0.19 +/- 0.006 | 0.38 +/- 0.010   | 0.74 +/- 0.025 | 0.40 +/- 0.013 | 0.31 +/- 0.011 | 0.23 +/- 0.010 | 0.14 +/- 0.014 |
>
>
> These new results show that the neural predictivity scores substantially improve when taking the noise ceiling into account. We thank R3 for pointing this out, and helping us put the original finding in perspective.
>
> > “There are some related work, e.g. https://www.science.org/doi/abs/10.1126/science.1245994, that could be mentioned in the paper.”
>
> We thank Reviewer 3 for this suggestion. This important study should certainly have been cited. We now indicate:
>
> ```“These elements, consistent with previous electrophysiological studies (Mesgarani & Chang Science 2014), provide a coherent spectrum of evidence for the location of acquired speech representations in the brain.”```
>
> (Note that other studies from Eddie Chang and Nima Mesgarani were cited.)
>
> > “Although the results are promising, please make the conclusion linked to the current set up and do not strongly generalize it to all cases.”
>
> We agree that some of our conclusions can be qualified. We now propose to add the following paragraph to the discussion:
>
> ```“It is important to stress that the scope of the present study could be broadened in several ways. First, our study focuses on adult speakers, whose cultural and educational background is not representative of the population (Henrich et al, 2010). Second, we focus on the passive listening of three languages. Third, we focus on one self-supervised learning architecture (Baevski et al. 2021), and its functional alignment with fMRI, whose temporal resolution is notoriously limited. Generalizing the present approach to more languages (Malik-Moraleda et al, 2022), a larger spectrum of children and adult participants recorded with a variety of electrophysiological and neuroimaging devices will thus be essential to confirm, precise, and/or mitigate the present findings.”```

---

### Official Review · Reviewer_6WfS · 2022-07-12

**Rating:** 5
**Confidence:** 5
**Soundness:** 2 fair
**Presentation:** 3 good
**Contribution:** 2 fair

**Summary:**

In this work, the authors build speech encoding models using wav2vec 2.0 and experiment with 4 different regimens:
1. Untrained network
2. Network trained on non-speech sounds
3. Network trained on non-native speech
4. Network trained on native speech

Their main findings are:
1. Representations from wav2vec 2.0 predict brain activity well in several regions. Although significantly different, the untrained networks have comparable effect sizes to the trained models.
2. Diff regions across cortex are best predicted by diff layers of the model such that low-level regions like the auditory cortex are best predicted by the lower layers.
3. Much like humans, the models do better at a phoneme discrimination task on the native language than non-native. The non-speech networks do worse, followed by the untrained networks.

**Questions:**

1. Selectivity for deeper layers in primary AC: In figure 3, why is primary auditory (or what is possibly Heschl’s gyrus) marked in red, ie., selective for deep layers? It would be useful to visualize the correlations for different layers in a few voxels to see if the curve is relatively flat and if this is an artifact of using a circular mean.
2. What do the individual participant maps look like for figure 3? Is the group-averaging introducing smoothness in the gradients that is not visible in individual maps ( due to the circular mean estimate being noisy)?
3. How were models trained for the ABX task? What are the specifications of the dataset and the task? Which layers were used to do this?
4. ABX task results: I imagine that the authors ran clustering for the English/French/non-speech data independently to get different quantization targets for each. Given that these targets roughly correspond to phonemes, is it surprising that the English model was better at distinguishing English phonemes than French? Similarly, is it surprising that English models outperform non-speech or untrained models that don’t have English quantization targets or have not learned how to use them respectively?
5. In Fig. 4D, it is surprising that only auditory cortex seems to care about language/speech specific features although we know that a large part of cortex processes natural speech. Can the authors comment on why this might be and perhaps visualize the individual correlations for all 5 conditions in a few example voxels per color?
6. What is the amount of data per participant in each language & stimulus set? Iiic, the Enligsh datasets have far more data per participant than French. How does the paper circumvent differences in the quality of model fits due to different amount of data being used? (especially for cross-language comparisons)

**Limitations:**

-NA-

**Strengths And Weaknesses:**

Edit after rebuttal: I comment the authors efforts to do a new analyses and the thorough responses to the review. Consequently, I have increased my score by 1. However, at this point I cannot recommend the paper for acceptance as some things stand out to me (more detail in follow-up comments):
- There is no established link between the different types of models tested, the layer-wise selectivity and the probing result. To this end, it is hard to interpret what the posited function of each region is or,
- what new things did we learn about the brain/deep speech networks that wasn't know previously? The paper seems largely confirmatory and the methods employed are not novel.
- some of the prediction correlations are surprisingly small even in areas robustly engaged during speech perception. Furthermore, some of the claims on selectivity for native vs. non-native vs. non-speech info is based on extremely small, possibly insignificant differences. This make interpretation hard.

###################################################################################################

Strengths:
1. Clarity: The paper is well written and easy to read.

Weaknesses:
1. Significance/Originality: The central premise of this paper lies in exploring self-supervised architectures for building encoding models. The results presented include 1) encoding model performance maps 2) layer wise selectivity maps 3) encoding performance viewed through a cascade of model selectivity (native > non-native....> chance). However, due to several reasons highlighted in detail below (like no discussion/analysis on what information is represented at different layers and how does this explain patterns of variation in layer selectivity, for example) and due to the identical results+analyses in prior work on supervised models and concurrent work on self-supervised models, the novelty of this work is not clear.
- Kell et al., Neuron 2018 -  exploring layer selectivity for diff supervised task architectures + specificity for speech vs. non-speech
- Millet et al., arXiv 2021 - exploring many of the analyses presented here but with supervised models
- Vaidya et al., ICML 2022 - exploring 4 self-supervised models, layer selectivity and probing to infer information at each layer
2. Under-explored/Limited analyses: Although the paper presents encoding model performance in every major analysis, the trends observed are not discussed further. For example, what information do different layers capture that could drive selectivity for lower layers in low-level brain regions vs. selectivity for deeper layers in high-level regions? Does this information emerge as a consequence of training? Does it emerge only when models are trained on speech? How about specificity of information for the native language? Without digging deeper into these questions, I am not sure of the significance of the results or how they inform our understanding of both speech models and the brain.
3. Overstating claims:
- Claims of supervision or lack thereof that match human ability:
    - Firstly, while the effect sizes are significantly different, these differences are still considerably small between the random/non-speech/non-native/native models.
    -  It is interesting to compare supervised vs. unsupervised models in terms of the information they capture and consequently, their ability to predict brain data. However:
        - given the implausibility of the learning mechanisms employed in these networks (as noted in the paper too; Lines 301-312)
        - the fact that they do not follow the same patterns of errors/do not mimic human behavior
        - the fact that the paper currently does not explore the types of information captured (what makes one model perform better than the other? how is this related to the purported functions of different regions?)

        , why is this a realistic model of speech perception? I agree that the model uses far less training data than LMs (which, contrary to how it is stated in the paper, are self-supervised too) or supervised speech models. However, without a serious/exhaustive analysis on the types of speech tasks these models do well on, there ability to mimic human speech errors etc., is it fair to state that we are moving towards brain-like models? (more detailed arguments in Guest & Martin, 2021) For example, lines 259-261: “Here, we test whether self-supervised learning on a limited amount of speech suffices to **yield a model functionally equivalent to speech perception in the human brain.** “

- in lines 290-300, the authors discuss several studies on the functional specificity of different brain regions during speech processing, functional gradients in PFC etc.. However, how is this being inferred form the data? I may have missed the link but iiuc this is not supported by the evidence presented.
- "the functional hierarchy of its transformer layers aligns with the cortical hierarchy of speech in the brain, and **reveals the whole-brain organisation of speech processing with an unprecedented clarity**" - Why is the layer wise selectivity important? Not to belabour the emphasis on what is captured by these representations, but without a discussion/analysis to reason why these differences might arise across regions, I am not sure what to make of these results or how they clarify the organization of speech processing across cortex. Furthermore, it appears that the upper-middle layers have high correlations in several regions across cortex, albeit not the highest. What do the authors make of this?
- Please cite related work appropriately. The authors have missed several relevant citations from PIs like Jack Gallant, Leila Wehbe, Alex Huth, Tom Mitchell, Christopher Honey, Jonathan Brennan, Alona Fyshe and more. For example, papers that were the first to build language encoding models or present the style of analyses reported here are missing.

---

> ### Author Response · Authors · 2022-08-02
> **Part 3A: clarification of the layer-preference for main regions-of-interest**
>
> > Question 1: Selectivity for deeper layers in primary AC: In figure 3, why is primary auditory (or what is possibly Heschl’s gyrus) marked in red, ie., selective for deep layers? It would be useful to visualize the correlations for different layers in a few voxels to see if the curve is relatively flat and if this is an artifact of using a circular mean.
>
>
> As indicated in Figure 3A-C, the primary auditory cortices are primarily correlated with early (not deep) layers, as expected. In supplementary materials, we now report the neural predictivity score for each layer, averaged across subjects, for two brain regions associated with acoustic processing (Heschl’s gyrus and sulcus), and three regions associated with higher level language processing (IFG) (Figure S4, Table below). While in Heschl, the voxels are mainly tund to the first transformer layer, in IFG, the voxels are mainly tuned to the  seventh and eighth layer of the transformer. For clarity, we here provide these results across the main regions of interest.
>
>
> |         | Heschl G             | Heschl S             | IFG Tri              | IFG Op               | IFG Orb              |
> |:--------|:---------------------|:---------------------|:---------------------|:---------------------|:---------------------|
> | Conv. 1 | 0.090 +/- 0.0034     | 0.094 +/- 0.0040     | 0.012 +/- 0.0012     | 0.013 +/- 0.0011     | 0.012 +/- 0.0013     |
> | Conv. 2 | 0.103 +/- 0.0043     | 0.108 +/- 0.0049     | 0.018 +/- 0.0013     | 0.017 +/- 0.0011     | 0.015 +/- 0.0015     |
> | Conv. 3 | 0.105 +/- 0.0044     | 0.110 +/- 0.0050     | 0.020 +/- 0.0014     | 0.019 +/- 0.0012     | 0.018 +/- 0.0016     |
> | Conv. 4 | 0.103 +/- 0.0045     | 0.110 +/- 0.0049     | 0.019 +/- 0.0014     | 0.018 +/- 0.0011     | 0.019 +/- 0.0015     |
> | Conv. 5 | 0.112 +/- 0.0043     | 0.115 +/- 0.0048     | 0.023 +/- 0.0014     | 0.021 +/- 0.0011     | 0.023 +/- 0.0014     |
> | Conv. 6 | 0.093 +/- 0.0040     | 0.100 +/- 0.0044     | 0.019 +/- 0.0013     | 0.017 +/- 0.0010     | 0.019 +/- 0.0014     |
> | Conv. 7 | 0.109 +/- 0.0039     | 0.113 +/- 0.0043     | 0.023 +/- 0.0014     | 0.020 +/- 0.0011     | 0.022 +/- 0.0014     |
> | Tr. 1   | **0.184 +/- 0.0057** | **0.190 +/- 0.0068** | 0.029 +/- 0.0016     | 0.027 +/- 0.0013     | 0.028 +/- 0.0015     |
> | Tr. 2   | 0.183 +/- 0.0056     | 0.189 +/- 0.0066     | 0.032 +/- 0.0015     | 0.027 +/- 0.0013     | 0.029 +/- 0.0016     |
> | Tr. 3   | 0.182 +/- 0.0057     | 0.188 +/- 0.0067     | 0.034 +/- 0.0016     | 0.028 +/- 0.0013     | 0.030 +/- 0.0016     |
> | Tr. 4   | 0.181 +/- 0.0058     | 0.186 +/- 0.0067     | 0.034 +/- 0.0016     | 0.028 +/- 0.0013     | 0.030 +/- 0.0017     |
> | Tr. 5   | 0.179 +/- 0.0058     | 0.183 +/- 0.0068     | 0.035 +/- 0.0016     | 0.029 +/- 0.0013     | 0.031 +/- 0.0017     |
> | Tr. 6   | 0.176 +/- 0.0056     | 0.181 +/- 0.0066     | 0.038 +/- 0.0017     | 0.031 +/- 0.0013     | 0.032 +/- 0.0016     |
> | Tr. 7   | 0.174 +/- 0.0057     | 0.181 +/- 0.0067     | **0.038 +/- 0.0017** | **0.031 +/- 0.0013** | 0.033 +/- 0.0017     |
> | Tr. 8   | 0.176 +/- 0.0057     | 0.183 +/- 0.0066     | 0.037 +/- 0.0017     | 0.031 +/- 0.0013     | **0.033 +/- 0.0018** |
> | Tr. 9   | 0.173 +/- 0.0056     | 0.180 +/- 0.0066     | 0.036 +/- 0.0017     | 0.030 +/- 0.0013     | 0.033 +/- 0.0017     |
> | Tr. 10  | 0.157 +/- 0.0053     | 0.160 +/- 0.0061     | 0.031 +/- 0.0017     | 0.027 +/- 0.0013     | 0.029 +/- 0.0017     |
> | Tr. 11  | 0.120 +/- 0.0046     | 0.122 +/- 0.0054     | 0.024 +/- 0.0014     | 0.022 +/- 0.0012     | 0.023 +/- 0.0015     |
> | Tr. 12  | 0.118 +/- 0.0043     | 0.119 +/- 0.0049     | 0.024 +/- 0.0015     | 0.021 +/- 0.0012     | 0.024 +/- 0.0016     |
>
> **Table F**: Neural predictivity scores, averaged across voxels, for each layer of the unsupervised Wav2Vec 2.0 model, on average across all voxels, and for five regions of interest. See Figures 3C and S4 for additional results.
>
>
>
> > Question 2: What do the individual participant maps look like for figure 3? Is the group-averaging introducing smoothness in the gradients that is not visible in individual maps ( due to the circular mean estimate being noisy)?
>
>
> Reviewer 2 is correct that the population-level analysis can introduce smoothness that would not be present at the individual subject level, as indicated in our discussion: *“it should be noted, however, that our results aggregate a large cohort of individuals which could mask a more modular organization at the individual level."*
>
> This consideration is not specific to the circular mean, which is here used to limit a regression-to-the mean-artifact.
> We believe that the study of inter-individual variability is outside the scope of the present research, and require the development of advanced methods based on tiling (Huth et al Nature 2016) and/or optimal transport (Thual et al, arXiv 2022).

---

> > ### Author Response · Authors · 2022-08-02
> > **Part 3B**
> >
> >
> > > Question 3: How were models trained for the ABX task? What are the specifications of the dataset and the task? Which layers were used to do this?”
> >
> > The ABX task does not require additional training. We agree with Reviewer 2 that the description of section 2.4 could be improved. We will indicate in the extended revised version of the manuscript (using the extra page allowed after the discussion period):
> >
> > *“To compare the phonetic representations of our models to those of humans, we use the method of Schatz et al 2016 and forced-choice discrimination judgements evaluated by online participants and publicly available (CC0: https://docs.cognitive-ml.fr/perceptimatic/).
> > Specifically, for each triplet, humans had to judge whether the stimulus X was closer to A or B. Analogously, we computed the Euclidean distance in the most discriminative layer (here transformer layer 5) to determine whether X was closer to A or B. Additional data, analyses and model-human comparison can be found in (Millet and Dunbar 2022).”*
> >
> >
> > > Question 4: ABX task results: I imagine that the authors ran clustering for the English/French/non-speech data independently to get different quantization targets for each. Given that these targets roughly correspond to phonemes, is it surprising that the English model was better at distinguishing English phonemes than French? Similarly, is it surprising that English models outperform non-speech or untrained models that don’t have English quantization targets or have not learned how to use them respectively?
> >
> >
> > This is correct; all models are trained independently and thus end up with different quantized targets. These quantized targets, however, are much more numerous and substantially shorter than phonemes, and insufficient research has been done to establish how well-aligned they are with phonemes in general; analyses of a model trained on English by Baevski et al. 2021 (Appendix D) suggested an imperfect relationship to phonemes. Precisely, they suggest that English models better distinguish English phonemes than French ones, because the models are optimized to capture the statistics of speech signals in the respective languages, including sub-phonemic and suprasegmental properties. Here, we formally test this prediction - It is also interesting to highlight that random, and non-speech models (which also have a quantization module) actually perform quite well – a result that is consistent with the fact that these models account for a fairly large portion of the brain responses to speech.
> >
> >
> > > Question 5: In Fig. 4D, it is surprising that only auditory cortex seems to care about language/speech specific features although we know that a large part of cortex processes natural speech. Can the authors comment on why this might be and perhaps visualize the individual correlations for all 5 conditions in a few example voxels per color?
> >
> > We agree that this point deserves additional comments, and thus add two supplementary tables with the individual correlation for all of the five conditions, with (Table S1) or without noise ceiling normalization (Table S2). We also propose to add the following paragraph to the Results section (in the extended version of the revised manuscript):
> >
> > *“Our analyses suggest that the native speech features are represented relatively high in the auditory processing hierarchy, i.e. in the superior temporal sulcus and gyrus (STS/G). This suggests that these regions are uniquely sensitive to the fine auditory structures specific to each language, and that higher level regions, such as the prefrontal and parietal cortices, may be not or less tuned to them.
> > We speculate that this new result fits the classical view of modular processing in the cortex (e.g. Fodor, 1983). For example, the Global Workspace Theory (Dehaene & Changeux 2011) predicts that higher level regions (PFC) would only be input with the result of sensory hierarchy, and would thus be blind to the many of its lower levels.”*

---

> > > ### Author Response · Authors · 2022-08-02
> > > **Part 3C: Neural predictivity scores corrected by the noise ceiling**
> > >
> > > |                      | Avg    | Top10   | Heschl   | STG    | STS    | IFG    | Motor   |
> > > |:---------------------|:-------|:--------|:---------|:-------|:-------|:-------|:--------|
> > > | Random Wav2Vec 2.0      | 13.9%  | 29.0%   | 66.9%    | 32.0%  | 21.8%  | 15.9%  | 11.9%   |
> > > | Non-Speech           | 16.4%  | 33.9%   | 71.0%    | 36.8%  | 26.9%  | 19.0%  | 11.7%   |
> > > | Non-Native           | 17.6%  | 35.9%   | 73.0%    | 39.0%  | 29.1%  | 21.0%  | 12.9%   |
> > > | Native, Supervised   | 18.3%  | 36.7%   | 74.2%    | 39.6%  | 29.8%  | 21.2%  | 13.6%   |
> > > | Native, Unsupervised | 18.8%  | 37.9%   | 74.4%    | 40.3%  | 31.3%  | 22.8%  | 13.8%   |
> > > | Noise ceiling        | 100.0% | 100.0%  | 100.0%   | 100.0% | 100.0% | 100.0% | 100.0%  |
> > >
> > > **Table S1. Neural Predictivity with noise ceiling normalization.** Neural predictivity divided by the noise ceiling, for the Narrative dataset, on average across all voxels (`Average`), for the 10% best voxels of the noise ceiling (`Top10`, Figure S2) and the voxels of five regions of interests.
> > >
> > > |                      |    Avg |   Top10 |   Heschl |    STG |    STS |    IFG |   Motor |
> > > |:---------------------|-------:|--------:|---------:|-------:|-------:|-------:|--------:|
> > > | Random Wav2Vec 2.0      | 0.0189 |  0.0693 |   0.1921 | 0.0712 | 0.0443 | 0.0237 |  0.0113 |
> > > | Non-Speech           | 0.0223 |  0.0804 |   0.2046 | 0.0814 | 0.0545 | 0.0283 |  0.0112 |
> > > | Non-Native           | 0.0239 |  0.0849 |   0.2106 | 0.0859 | 0.0589 | 0.0315 |  0.0123 |
> > > | Native, Supervised   | 0.0247 |  0.0865 |   0.2133 | 0.0871 | 0.0602 | 0.0318 |  0.013  |
> > > | Native, Unsupervised | 0.0254 |  0.0893 |   0.2144 | 0.0886 | 0.0635 | 0.0343 |  0.0131 |
> > > | Noise ceiling        | 0.1174 |  0.219  |   0.2873 | 0.1808 | 0.1961 | 0.149  |  0.0938 |
> > >
> > > **Table S2. Neural Predictivity without noise ceiling normalization.** Same as Table S1, but without dividing by the noise ceiling estimates.
> > >
> > > > Question 6: What is the amount of data per participant in each language & stimulus set? Iiic, the English datasets have far more data per participant than French. How does the paper circumvent differences in the quality of model fits due to different amount of data being used? (especially for cross-language comparisons)
> > >
> > > The 48 English, 33 Mandarin and 28 French participants who listened to the Little Prince were scanned during 94, 90 and 97 minutes of effective speech, respectively.
> > >
> > > In addition, we will specify:
> > > *“The 303 [English] participants [from the Narrative datasets] listened different subset of the audio, from 7 to 98 min of fMRI data with speech (mean=26min)”*
> > >
> > > All Wav2Vec models are trained with the same amount of data (600 hours). The mapping between a Wav2Vec 2.0 model and the brain is done at the *participant* level (i.e. independently of all other participants). The cross-language comparisons are applied within each participant: i.e. For each English subject, w2v_english is compared to two non-native models: w2v_mandarin and w2v_french; and we use second-level statistics across participants to report the summary metrics.

---

> ### Author Response · Authors · 2022-08-02
> **Part 2A: Effect sizes**
>
> > “Claims of supervision or lack thereof that match human ability:
> > - Firstly, while the effect sizes are significantly different, these differences are still considerably small between the random/non-speech/non-native/native models.”
>
> Reviewer 2 is right that the present effect sizes are relatively small.
>
> First, the order of magnitude of the main effects are similar to those of previous studies from multiple groups (Huth et al Nature 2016, Jain & Huth Neurips 2018, Toneva & Wehbe Neurips 2019, Caucheteux & King, Nature Comm 2022). The low effect sizes are expected given that we here try to model individual voxels at the single TR level, which is notoriously noisy. For clarity, we have now performed a noise-ceiling analysis, and also report our results in proportion to what this analysis estimates to be explainable. The new results suggest that wav2vec 2.0 explains 19% of the noise ceiling on average across the whole cortex, and up to 74% in Heschl’s gyrus.
>
> Second, many of the effects reported in our study are averaged across all cortical voxels, for simplicity and to avoid cherry picking regions of interest. Because many voxels do not respond much to speech (Malik-Moraleda et al, Nature Neuroscience 2022), these averages expectedly become very small. For clarity, we add a supplementary table for each critical comparison (non–speech vs random, non-native vs non-speech and native vs non-native). The results show that the relative improvements can peak to 26%, 13% and 9% in IFG, for the non-speech, non-native and native models, respectively (Table E).
>
> | Model A    | Model B    | Heschl     | STG         | STS         | IFG         |
> |:-----------|:-----------|:-----------|:------------|:------------|:------------|
> | Non speech | Random     | 6% +/- 0.8 | 25% +/- 6.0 | 27% +/- 3.9 | 26% +/- 5.0 |
> | Non native | Non speech | 3% +/- 0.4 | 14% +/- 6.0 | 9% +/- 1.7  | 13% +/- 2.4 |
> | Native     | Non native | 2% +/- 0.3 | 5% +/- 2.1  | 9% +/- 1.3  | 9% +/- 1.9  |
>
> **Table E.** Relative improvement of the neural predictivity scores for each model pair. Precisely, for each pair of models (model A, model B), we compute the relative improvement in neural predictivity scores of model A over model B (A-B/B). Scores are averaged across subjects, and voxels within each regions of interest.

---

> > ### Author Response · Authors · 2022-08-02
> > **Part2B: Clarification of the implications of the present results**
> >
> >
> > > - “It is interesting to compare supervised vs. unsupervised models in terms of the information they capture and consequently, their ability to predict brain data. However:
> > > (i) given the implausibility of the learning mechanisms employed in these networks (as noted in the paper too; Lines 301-312)
> > > (ii) the fact that they do not follow the same patterns of errors/do not mimic human behavior
> > > (iii) the fact that the paper currently does not explore the types of information captured (what makes one model perform better than the other? how is this related to the purported functions of different regions?)
> > > , why is this a realistic model of speech perception? [...]  is it fair to state that we are moving towards brain-like models? (more detailed arguments in Guest & Martin, 2021) For example, lines 259-261: “Here, we test whether self-supervised learning on a limited amount of speech suffices to yield a model functionally equivalent to speech perception in the human brain. “
> >
> > We agree that the goal expressed in this last sentence may be excessive, and that our discussion on the remaining gaps between wav2vec 2.0 and the brain could be improved. We will indicate in the extended version of the revised manuscript:
> >
> > *“Here, we test whether self-supervised learning on a limited amount of speech effectively accounts for the organization of speech processing in the human brain as measured with fMRI.”*
> >
> > Furthermore, we propose to amend the discussion as follows:
> >
> >
> > *“Our study shows that a self-supervised model, trained on a remarkably small amount of unlabelled data, effectively accounts for cortical responses to speech. This learning scheme is considerably more realistic than the classic supervised models, thus promises to help understand how the human brain learns and organizes speech processing.*
> >
> > *However, these results do not imply that wav2vec 2.0 has learned all of the speech representations used by the brain. Indeed, several major gaps remain between self-supervised speech models like Wav2Vec 2.0 and the brain. First, its transformer layers are not temporally constrained: each layer can access all elements within the contextual window. This differs from the necessarily recurrent nature of processing in the brain. Second, Wav2Vec 2.0 shows differences with humans in recent behavioral studies: it shows a higher sensitivity to band-pass filtering and an under-reliance on fine temporal structures (Weerts et al. 2021). It also fails to predict categorical effects on perception (Millet et al. 2021). Third, recent studies show that Wav2Vec 2.0 encodes significantly less semantic information than text-based models (Pasad et al. 2021) (Vaidya, Jain & Huth 2022). Our analyses suggest that learning allows Wav2Vec 2.0 to capture some lexical features in its deep layers (Figure S1). However, whether these layers also whether they capture complex syntactic structures, such as recursive syntactic trees, remains an open question (Lakrertz et al 2022). We speculate that these limitations may be due to the time scales of Wav2Vec 2.0 which, unlike humans, learns very short-term representations of speech.*
> >
> > *Overall, these differences likely explain why the neural predictivity scores of Wav2Vec 2.0 remain substantially lower than our noise-ceiling (19% on average, and up to 74% in Heschl Gyrus, Table S1). Together with epistemological accounts (Guest and Martin 2021) and concurrent results (Vaidya, Jain & Huth 2022), our results highlight the remaining gap between current self-supervised models and the human brain.”*

---

> > > ### Author Response · Authors · 2022-08-02
> > > **Part 2C**
> > >
> > >
> > > > “in lines 290-300, the authors discuss several studies on the functional specificity of different brain regions during speech processing, functional gradients in PFC etc.. However, how is this being inferred for the data? I may have missed the link but iiuc this is not supported by the evidence presented.”
> > >
> > > The functional specificity corresponds to the voxel-wise comparison between random, non-speech, non-native and native models described in Figure 4. The functional gradients correspond to the voxel-wise analysis of layer specificity displayed in Figure 3.
> > > We now added the references to the figure in the corresponding discussion paragraph.
> > >
> > >
> > > > “the functional hierarchy of its transformer layers aligns with the cortical hierarchy of speech in the brain, and reveals the whole-brain organisation of speech processing with an unprecedented clarity" - Why is the layer wise selectivity important? “
> > >
> > > This result suggests that self-supervised learning suffices to generate a hierarchy of representations similar to the brain’s. The layer-wise selectivity is a simple method to reveal this hierarchical organization. The concurrent submission of Vaidya, Jain & Huth (2022) reaches a similar conclusion – although with significantly noisier cortical maps – using an alternative method based on the interpretation of PCA coefficients.
> > >
> > > Finally, and as detailed above, we now clarify the information encoded in each layer of wav2vec 2.0, and show that those specific to language are encoded in the middle and deep layers.
> > >
> > > > “Furthermore, it appears that the upper-middle layers have high correlations in several regions across cortex, albeit not the highest. What do the authors make of this?”
> > >
> > > This is an interesting remark. We will add a discussion of the features encoded in each region. In particular:
> > >
> > > *“Interestingly, the word and sentence-level features are encoded deeper in the supervised network (best layer=18 in Table 5) compared to the unsupervised network (best layer=14), which suggests that self-supervised learning generates a reservoir representations in its middle layers, reservoir which may partly overlap with the labels used in supervised learning.”*

---

> ### Author Response · Authors · 2022-08-02
> **Part 1A: Comparison to recent studies**
>
> We thank Reviewer 2 for their thorough review, as well as for their these valuable questions and remarks, which we now address below.
>
> > Weaknesses:
> > 1\. Significance/Originality: The central premise of this paper lies in exploring self-supervised architectures for building encoding models. The results presented include 1) encoding model performance maps 2) layer wise selectivity maps 3) encoding performance viewed through a cascade of model selectivity (native > non-native....> chance). However, due to several reasons highlighted in detail below (like no discussion/analysis on what information is represented at different layers and how does this explain patterns of variation in layer selectivity, for example) and due to the identical results+analyses in prior work on supervised models and concurrent work on self-supervised models, the novelty of this work is not clear.
> > - Kell et al., Neuron 2018 - exploring layer selectivity for diff supervised task architectures + specificity for speech vs. non-speech
> > - Millet et al., arXiv 2021 - exploring many of the analyses presented here but with supervised models
> > - Vaidya, Jain & Huth., ICML 2022 - exploring 4 self-supervised models, layer selectivity and probing to infer information at each layer
>
> We agree that these three studies are relevant to the present work, and indeed devote a discussion paragraph to the supervised convnets studied in Kell et al (Neuron 2018) and Millet & King (arXiv 2021).
>
> For context, Millet & King (arXiv 2021) is a preprint that was not accepted; we here capitalize on the criticisms it received on OpenReview to build the present study.
>
> Furthermore, Vaidya, Jain & Huth (arXiv 2022) was released after the present submission. According to the [Neurips guidelines](https://neurips.cc/Conferences/2022/PaperInformation/NeurIPS-FAQ), this study, which share several of our conclusions, cannot be used to discard the novelty of our work: “​​Papers appearing less than two months before the submission deadline are generally considered concurrent to NeurIPS submissions. Authors are not expected to compare to work that appeared only a month or two before the deadline.”
>
> For completeness, we will nevertheless clarify how the present paper makes significant contributions beyond these three studies.

---

> > ### Author Response · Authors · 2022-08-02
> > **Part 1B: Comparison to Vaidya, Jain & Huth arXiv 2022**
> >
> > Our objective is here to investigate whether a biologically plausible learning mechanism – namely self-supervised learning on a small amount of data – suffices to account for the functional organization of speech processing learnt in the brain.
> >
> > ## Approach
> >
> > For this, we analyze the fMRI of n=412 participants (compared to n=6 for Vaidya, Jain & Huth and n=8 for Kell et al) from n=3 different native languages (compared to n=1 in previous encoding studies). We compare these brain responses to those of different models, all trained with the same architecture but with different objectives (supervised vs non-supervised) and different datasets (600 hours of either non-speech, non-native or native speech sounds). By contrast, neither Kell et al nor Millet & King study self-supervised models, and Vaidya, Jain & Huth did not evaluate how different training led the same architecture to be more-or-less similar to the brain.
> >
> > ## Contributions
> >
> > The strength of this approach (considerably more data, minimal comparison between models) allow us to reveal three main results:
> >
> > * Result 1: Self supervision leads the model to be more similar to the brain than supervision. Unlike Vaidya, Jain & Huth (arXiv 2022), we make this comparison with the same architecture, and the same amount of data, and can thus be confident that it is the learning objective that proved important.
> >
> > * Result 2: The hierarchy of this self-supervised model is similar to the brain’s. To our knowledge, this hierarchy analysis leads to much clearer results than previous studies. First, we reveal, for each voxel of the whole cortex, its preferred layer in the model. By contrast, Kell et al (2018) only investigate the preference for middle versus deep layers within the temporal lobe; Vaidya, Jain & Huth explore a similar issue and draw similar conclusions to ours by inspecting the loading of a PCA. Second, we discover new hierarchical gradients, e.g. within the prefrontal cortex, a region tightly linked to human cognition, learning and control and whose functional organization remains largely unknown (Fuster, 2015).
> >
> > * Result 3: For the first time, we provide empirical evidence of language-specific representations of speech in the brain: i.e. our self-supervised models best align with the brain when they have been exposed to the native language of the participants. As previous studies were monolingual, this language_model x language_participants comparison could not be tested.
> >
> > Overall, we believe that Vaidya, Jain & Huth’s study is strong and corroborates several of our findings. Despite the fact that this concurrent study was put on arXiv after the conference submission deadline, we agree that our manuscript would benefit from adding the above discussion to the manuscript.
> >
> >
> >
> > |                                     | Kell et al      | Millet & King    | Vaidya, Jain & Huth    | Ours             |
> > |-------------------------------------|-----------------|------------------|-----------------|------------------|
> > | Released/Accepted before submission | yes             | no               | no              |                  |
> > | # Subjects                          | 8               | 102              | 6               | 417              |
> > | # Languages                         | 1               | 1                | 1               | 3                |
> > | Scope                               | Temporal lobe   | Full brain       | Full brain      | Full brain       |
> > | Stimuli                             | Auditory scenes | Speech           | Speech          | Speech           |
> > | Models                              | Supervised      | Supervised       | Self-supervised | Self-supervised  |
> > | Comparison                          | Architectures   | Learning schemes | Architectures   | Learning schemes |
> >
> >
> > > Please cite related work appropriately. The authors have missed several relevant citations from PIs like Jack Gallant, Leila Wehbe, Alex Huth, Tom Mitchell, Christopher Honey, Jonathan Brennan, Alona Fyshe and more. For example, papers that were the first to build language encoding models or present the style of analyses reported here are missing.
> >
> > We thank Reviewer 2 for pointing this out. Note that most of these authors are cited: Jack Gallant (n=3), Leila Wehbe (n=2), Alex Huth (n=2), Christopher Honey (n=2), Jonathan Brennan (n=1).
> >
> > Nevertheless, we agree that several seminal papers are missing. We thus added Mitchell et al (Science 2008), Wehbe & Mitchell et al (PLoS One 2014), Jain & Huth (Neurips 2018) and Jain, Huth et al (Neurips 2020) and well as the recent Vaidya, Jain & Huth preprint.

---

> > > ### Author Response · Authors · 2022-08-02
> > > **Part 1C: requested interpretation analyses of the representation in each layer of wav2vec 2.0**
> > >
> > > > 1\. “Under-explored/Limited analyses: Although the paper presents encoding model performance in every major analysis, the trends observed are not discussed further. For example, what information do different layers capture that could drive selectivity for lower layers in low-level brain regions vs. selectivity for deeper layers in high-level regions? “
> > > > “Does this information emerge as a consequence of training? “
> > > > “Does it emerge only when models are trained on speech?”
> > > > “How about specificity of information for the native language?”
> > >
> > > Our objective is to test whether (not why) an unsupervised model of speech processing in the brain can effectively account for its functional organization as measured with fMRI. However, we agree that interpretation is an interesting topic.
> > >
> > > ### Results
> > > Following Reviewer 2’s remark, we now add a layer-wise inspection of our models of spectro-temporal, phonetic and word representations. We will add the following paragraph in the paper (using the additional page authorized after the discussion period):
> > >
> > > *“To investigate what features the different layers of Wav2Vec 2.0 encode, we perform a ridge regression on the [timit dataset](https://catalog.ldc.upenn.edu/LDC93S1) to predict five auditory and linguistic features from the activation functions of each layer and model of the present paper. We explore the following features:
> > > -  the MEL spectrogram of the audio, computed using librosa (McFee et al. 2015, d=128)
> > > - the phonemes (categorical features). We use the transcripts and alignments provided in Timit.
> > > - the word embedding and part-of-speech of the words. The time alignments for words are provided by Timit. We use spaCy to compute the word embedding (medium model, d=300), and their part-of-speech (categorical feature, d=19)
> > > the sentence embedding of each sample, provided by Laser.*
> > >
> > > *We use a subset of 1680 samples from Timit, each sample being an audio recording of a short sentence (<10 seconds) from 24 speakers. The model's activations were mean-pooled to the sampling rate of each feature.*
> > >
> > > *The results show that the layers of Wav2Vec 2.0 partially follows the hierarchy predicted from neuro-linguistics (Hickok and Poeppel, 2007) (Table D): the first layers of the transformer best account for the spectro-temporal information, whereas deeper layers best account for the phonetic, word-level and sentence level information. While all of these features emerge with training (Figure S1), only the highest-level features (phone, word and sentence-level) appear to be specific to speech and to the language with which wav2vec 2.0 was trained (Figure S1).”*
> > >
> > >
> > > ### Discussion
> > > *“Interpreting the representations of deep learning models is notoriously difficult, and the topic of dedicated studies. For example, Pasad et al. 2021 explored the encoding of local acoustic features, phone identity, word identity and word meaning across layers. Similarly, Millet, Chitoran and Dunbar (2021) compared representations to human behavioral data to assess whether they better captured listeners’ perception of higher-level phonemic properties or of lower-level subphonemic properties of speech stimuli. Finally, Vaidya, Jain & Huth’s recent study explores filter banks, spectrograms, phonemes and words across layers. Here, we complement these analyses by showing that self-supervised learning allows wav2vec 2.0 to learn represents, along its hierarchy the representations of MEL spectrograms, phonetic categories and word embeddings (Figure S1).*
> > >
> > > *"Critically, only the highest-level features, namely phonetic- and word- and sentence-level features appeared to be specific to (1) speech and (2) to the language with which it was trained (Figure S1). Interestingly, the word and sentence-level features are encoded deeper in the supervised network (best layer=18 in Table D) compared to the unsupervised network (best layer=14), which suggests that self-supervised learning generates a reservoir representations in its middle layers, reservoir which may partly overlap with the labels used in supervised learning. Together with our ABX tests, and layer-wise tuning of each voxel (Figure 3), these elements suggest that the representations of speech shaped by our experience are learnt and instantiated in the superior temporal gyrus and sulcus. These elements, consistent with previous electrophysiological studies (Mesgarani & Chang Science 2014), thus provide a coherent spectrum of evidence for the location of acquired speech representations in the brain.”*
> > >
> > > We thank Reviewer 2 for helping us improve this facet of our study.

---

> > > > ### Author Response · Authors · 2022-08-02
> > > > **Table D**
> > > >
> > > > |                               |   MEL |   Phone |   Word embedding |     POS |   Sentence embedding |   Avg |
> > > > |:------------------------------|------:|--------:|-----------------:|--------:|---------------------:|------:|
> > > > | Random Wav2Vec 2.0               |   2   |  8.7    |           8      |  8.9    |               8.1    |  7.14 |
> > > > | Acoustic Wav2Vec 2.0             |  12.5 | 15.7    |          14      | 14.4    |              14.2    | 14.16 |
> > > > | Mandarin Wav2Vec 2.0             |   9.1 | 11.9    |          12.2    | 11.9    |              13      | 11.62 |
> > > > | French Wav2Vec 2.0               |   8   | 11      |          12.7    | 11.8    |              13      | 11.3  |
> > > > | Dutch Wav2Vec 2.0                |  18.9 | 11.4    |          12      | 12.4    |              13      | 13.54 |
> > > > | English Wav2Vec 2.0              |   8   | 15.2    |          14      | 14.4    |              14      | 13.12 |
> > > > | English Wav2Vec 2.0 (supervised) |   8   | 16.9    |          18      | 18      |              18      | 15.78 |
> > > > | Avg                           |   9.5 | 12.9714 |          12.9857 | 13.1143 |              13.3286 | 12.38 |
> > > >
> > > > **Table D.** For each model (row) and target (column), the layer that maximizes probing performance (Figure S1), averaged across the 10 cross-validation folds. See Appendix for corresponding Figure.

---

> ### Comment · Reviewer_6WfS · 2022-08-09
> **Comments to part 1**
>
> > By contrast, neither Kell et al nor Millet & King study self-supervised models, and Vaidya, Jain & Huth did not evaluate how different training led the same architecture to be more-or-less similar to the brain.
>
> I agree with the above and the important point the authors noted that one of the studies was made public concurrently with the current work. However, I still maintain my comment that given a lot of recent studies on language encoding models and Kell et al. + Millet et al., is it really surprising that training deep networks leads to better brain prediction than a random network?
>
> > Self supervision leads the model to be more similar to the brain than supervision.
>
> > Our objective is to test whether (not why) an unsupervised model of speech processing in the brain can effectively account for its functional organization as measured with fMRI.
>
> Additionally, I understand that the aim was "to test whether (not why)" but hasn't this question been answered indirectly by the huge body of work showing the types of information these models capture and their ability to transfer to downstream tasks? And I appreciate the authors efforts at doing the probing analysis. However, what about self-supervision helps? How does it impact the types of information learned and consequently, the types of information used for down-stream prediction?
>
> > Second, we discover new hierarchical gradients, e.g. within the prefrontal cortex, a region tightly linked to human cognition, learning and control and whose functional organization remains largely unknown (Fuster, 2015).
>
> What gradients are being referred to here?
>
> > we provide empirical evidence of language-specific representations of speech in the brain
>
> What does this mean?
>
> I believe the authors have a very promising dataset and methodology but the results in their current form are not unexpected. Unfortunately, I do not think they tell us anything new about how the brain/deep networks function. For example, the speech hierarchy across cortex is known and has been shown with deep learning-based encoding models. The novelty here is to show the same thing with self-supervision which is unsurprising given other adjacent studies in both speech ML and language neuroscience.
>
> Re probing analysis: I greatly appreciate the author's efforts to run the probing analysis in such a short time. I will consequently increase my score by 1 point.
>
> Follow-up questions:
> 1. What does the performance look like across layers? I believe taking the max across layers could hide lack of any significant trends across layers.
> 2. The effect of training is a clean result and imo the authors should discuss the connection between this and prediction performance changes with training explicitly.
> 3. The authors do not address the large differences across models in probing performance, especially for Mel. Furthermore, other studies have shown larger gaps in layer-wise selectivity for features like Mel vs. word unlike the small range shown here (9-12).
> 4. I understand there were both space and time limitations but I do believe that the paper would greatly benefit from a way to link the probing results to the layer-preference maps. Currently, looking at them in tandem is not sufficient to understand what types of information each region could be representing. Especially, given the small range over which the averages vary per task.
> 5. Is TIMIT available in other language or are the authors using English TIMIT to test even the non-English models? Wouldn't that be a possible source of confound?
> 6. What are the accuracies achieved by the best layers per model and task?

---

> ### Comment · Reviewer_6WfS · 2022-08-09
> **Comments to part 2**
>
> I thank the reviewers for clarifying some points in the discussion and estimating the noise ceiling.
>
> Re noise ceiling: I think the authors have a potentially interesting setup here- is there a meaningful way to link the probing results across different types of models, the % of noise ceiling that's reached and layer-wise selectivity to better understand the differences between anatomical areas? I think the information is there but I'd be excited to see a more in-depth analysis of why the noise ceiling %s look so different across ROIs and how this relates to the types of information that arises in different networks.
>
> Re "new gradients in PFC": I am still not sure what the "new" gradients are. How are the layer wise preference maps revealing a new gradient? What does said gradient correspond to?
>
> Re the following (I believe it was misunderstood):
> > “Furthermore, it appears that the upper-middle layers have high correlations in several regions across cortex, albeit not the highest. What do the authors make of this?”
>
> I think this points to a possibility that the upper-middle layers encode both acoustic information as well as more abstract information like words, which allows them to predict so much of the cortex. I'd urge the authors to explore this.

---

> ### Comment · Reviewer_6WfS · 2022-08-09
> **Comments to part 3**
>
> Re the following which I believe was misunderstood:
> > In figure 3, why is primary auditory (or what is possibly Heschl’s gyrus) marked in red,
>
> There is a band of "red" regions overlapping the "blue" regions. Can the authors explain this?
>
> In the neural predictive scores table shown in the rebuttal, I'd urge the authors to either use a correlation threshold or significance test to select voxels per ROI. the correlations for IFG look suspiciously low- why is that?
>
> Re individual maps: I wanted to see the individual maps to gauge how noise the circular mean is as a metric for layer wise preference.
>
> Re different quantization targets across diff language models- I agree that one cannot consider the targets to be phonemes but an imperfect approximation. It is still unclear to me why Dutch models performing better than English at predicting Dutch speech elicited brain responses is surprising? Given what both the authors and I noted:
> > Precisely, they suggest that English models better distinguish English phonemes than French ones, because the models are optimized to capture the statistics of speech signals in the respective languages, including sub-phonemic and suprasegmental properties.
>
> Very interesting point, but why does this happen? how does it relate to the probing results?
> >  It is also interesting to highlight that random, and non-speech models (which also have a quantization module) actually perform quite well – a result that is consistent with the fact that these models account for a fairly large portion of the brain responses to speech.
>
> Before I interpret the results in Fig. 4 and tables S1,2:
> - Why are the neural predictive scores so small in every region but HG?!
> - The noise ceiling %s hide the fact that the actual differences in correlation are pretty small. For ex., 0.02 between random and native speech in HG. I'd urge the authors to test for significance!
>
> Re diff amounts of training data: Given the very strong effect of amount of training data on encoding model performance, I am confused why the authors report results that average across differently sized datasets. This could artificially deflate the performance across cortex or differences between layers, models etc..

---

> > ### Author Response · Authors · 2022-08-09
> > **2nd round to Reviewer 6WfS: Summary**
> >
> > # Summary
> > We thank Reviewer 2 for their detailed review.
> >
> > We note that the results of the analyses, namely (1) the noise ceiling, (2) the alternative models, and (3) the systematic probing (MEL, word and sentence-level representations $\times$ each layer of the model  $\times$ training type) requested by the Reviewer all strengthened our original conclusions.
> >
> > The main remaining concern of Reviewer 2 is on originality and interpretation:
> >
> > > I agree with the above and the important point the authors noted that one of the studies was made public concurrently with the current work. However, I still maintain my comment that given a lot of recent studies on language encoding models and Kell et al. + Millet et al., is it really surprising that training deep networks leads to better brain prediction than a random network?
> >
> > > Additionally, I understand that the aim was "to test whether (not why)" but hasn't this question been answered indirectly by the huge body of work showing the types of information these models capture and their ability to transfer to downstream tasks?
> >
> > > I believe the authors have a very promising dataset and methodology but the results in their current form are not unexpected. Unfortunately, I do not think they tell us anything new about how the brain/deep networks function. For example, the speech hierarchy across cortex is known and has been shown with deep learning-based encoding models.
> >
> > We appreciate these elemlents of concern. To limit a debate of opinions about what could have been expected, we list the main fact, as follows:
> >
> > First, our study is timely: SSL has only recently proved efficient in deep learning, and has thus not been studied in speech neuroscience. This novelty explains why our study finds similar conclusions to Vaydia, Jain & Huth's strong but concurrent submission.
> >
> > Second, the other studies indicated by Reviewer 2 are either (1) not focusing on spoken language (Kell et al, n=8 participants listening to snippets of various sounds) or (2) previously rejected, and thus, the very basis of the present study (Millet & King, n=102 participants).
> >
> > Third, while some elements of the speech hierarchy in the cortex are “known”, _the learning rules that allow the human brain to learn speech processing certainly remains one of the greatest challenges to cognitive and computational neuroscience_ (and linguistics, and A.I. for that matter).
> >
> > Here, we show, with unprecedentedly large amount of data (n=412 participants, 3 languages) that SSL brings us one step closer to solve this issue: with a plausible amount of speech (only 600 hours) and no supervision, this learning objective allows the learning of brain-like representations, organized according to a similar hierarchy, and tuned to the native language. None of these elements had been shown before.
> >
> > Our objective is not to solve and understand language in the brain, but to test whether a simple learning principle could suffice to explain its functional organization.
> >
> > We now turn to the more specific questions.

---

> > > ### Author Response · Authors · 2022-08-09
> > > **2nd round to Reviewer 6WfS: Part 3**
> > >
> > > > Re the following which I believe was misunderstood:
> > > > In figure 3, why is primary auditory (or what is possibly Heschl’s gyrus) marked in red, There is a band of "red" regions overlapping the "blue" regions. Can the authors explain this?
> > >
> > > The thin band of red voxels in this area disappears with a slightly more conservative threshold.
> > > We now propose to add:
> > >
> > > ```“Some voxels around the auditory cortices seem to be tuned to the highest layers of wav2vec 2.0. Additional research focusing on a higher resolution of this region remains necessary to disentangle the role that feedback may play to explain this unexpected phenomenon.”```
> > >
> > > > In the neural predictive scores table shown in the rebuttal, I'd urge the authors to either use a correlation threshold or significance test to select voxels per ROI. the correlations for IFG look suspiciously low- why is that?
> > >
> > > The correlations in IFG are relatively low because the level of noise is relatively high in IFG (Supplementary Figure S2). We are happy to add the table based on a correlation threshold.
> > >
> > > > Re individual maps: I wanted to see the individual maps to gauge how noise the circular mean is as a metric for layer wise preference.
> > >
> > > We provide in Supplementary Figure S4 the correlation scores for each layer, *before applying the circular mean*. Figure S4 shows that the argmax layer in IFG is deeper than the argmax layer in Heschl.
> > >
> > > > Re different quantization targets across diff language models- I agree that one cannot consider the targets to be phonemes but an imperfect approximation. It is still unclear to me why Dutch models performing better than English at predicting Dutch speech elicited brain responses is surprising? Given what both the authors and I noted: Precisely, they suggest that English models better distinguish English phonemes than French ones, because the models are optimized to capture the statistics of speech signals in the respective languages, including sub-phonemic and suprasegmental properties.
> > > > Very interesting point, but why does this happen? how does it relate to the probing results?
> > >
> > > The probing results confirm such observations: Supplementary Figure S1 shows that the English network has better probing performance for phoneme classification than the non-English networks. We agree that, ultimately, the goal is to build a model that automatically demonstrates these properties. The aim of this analysis is to test and quantify such emergence phenomenon, and show where the corresponding representations specifically map onto the brain.
> > >
> > > > It is also interesting to highlight that random, and non-speech models (which also have a quantization module) actually perform quite well – a result that is consistent with the fact that these models account for a fairly large portion of the brain responses to speech.
> > > Before I interpret the results in Fig. 4 and tables S1,2:
> > >
> > > > - Why are the neural predictive scores so small in every region but HG?!
> > >
> > > The absolute neural predictivity scores are small in regions as IFG because the level of noise is high in those regions (Supplementary Figure S2). The amount of explainable signal, though, is high compared to the level of noise (e.g. 23% in IFG).

---

> > > > ### Author Response · Authors · 2022-08-09
> > > > **2nd round to Reviewer 6WfS: Part 3 (ctd)**
> > > >
> > > >
> > > > > - The noise ceiling %s hide the fact that the actual differences in correlation are pretty small. For ex., 0.02 between random and native speech in HG. I'd urge the authors to test for significance!
> > > >
> > > > We agree. These effect sizes were originally reported in absolute values to avoid giving a false view of our results.
> > > >
> > > > While small, the differences between the networks are highly significant: between random and native speech, $p<10^{-46}$ (Figure 1), in HG specifically: $p<10^{-28}$.  We now report the p-values for each difference in representative regions:
> > > >
> > > > |                         | Avg        | Top10NoiseCeil   | Heschl     | STG        | STS        | IFG        | Motor     |
> > > > |:------------------------|:-----------|:-----------------|:-----------|:-----------|:-----------|:-----------|:----------|
> > > > | Non speech - Random     | $10^{-14}$ | $10^{-22}$       | $10^{-13}$ | $10^{-27}$ | $10^{-12}$ | $10^{-5}$  | n.s. |
> > > > | Non native - Non speech | $10^{-8}$  | $10^{-14}$       | $10^{-12}$ | $10^{-19}$ | $10^{-8}$  | $10^{-6}$  | n.s. |
> > > > | Native - Non native     | $10^{-10}$ | $10^{-15}$       | $10^{-7}$  | $10^{-11}$ | $10^{-11}$ | $10^{-5}$  | n.s. |
> > > > | Native - Random         | $10^{-31}$ | $10^{-41}$       | $10^{-28}$ | $10^{-42}$ | $10^{-32}$ | $10^{-17}$ | n.s. |
> > > >
> > > > **Table G**: Significance of the difference in neural predictivity scores between the networks (random, non-speech, non-native and native). Significance is assessed using a two-sided Wilcoxon test provided by Scipy, testing whether the difference is different from zero.
> > > >
> > > > > Re diff amounts of training data: Given the very strong effect of amount of training data on encoding model performance, I am confused why the authors report results that average across differently sized datasets. This could artificially deflate the performance across cortex or differences between layers, models etc.
> > > >
> > > > We do *not* fit encoding models across languages, datasets or subjects. One encoding model is fitted for each subject, on the dataset corresponding to the amount of speech to which they listen to while being recorded with fMRI. Thus, the amount of training data is comparable across encoding models.

---

> > > ### Author Response · Authors · 2022-08-09
> > > **2nd round to Reviewer 6WfS: Part 2**
> > >
> > > > I thank the reviewers for clarifying some points in the discussion and estimating the noise ceiling. Re noise ceiling: I think the authors have a potentially interesting setup here- is there a meaningful way to link the probing results across different types of models, the % of noise ceiling that's reached and layer-wise selectivity to better understand the differences between anatomical areas? I think the information is there but I'd be excited to see a more in-depth analysis of why the noise ceiling %s look so different across ROIs and how this relates to the types of information that arises in different networks. "new gradients in PFC": I am still not sure what the "new" gradients are. How are the layer wise preference maps revealing a new gradient? What does said gradient correspond to?
> > >
> > > We implemented the noise-ceiling analyses requested by the Reviewer as well as the several probing analyses. The results strengthened the interpretability of our findings, and did not impact our original conclusions. Why some regions have relatively large noise ceilings is an open question, which highlights the necessity to go beyond the present model and learning rule.
> > >
> > > We now respond to this point above. Specifically, we propose to add a supplementary cross-section to clarify the functional gradients identified in the prefrontal cortex.
> > >
> > > > Re the following (I believe it was misunderstood): “Furthermore, it appears that the upper-middle layers have high correlations in several regions across cortex, albeit not the highest. What do the authors make of this?”. I think this points to a possibility that the upper-middle layers encode both acoustic information as well as more abstract information like words, which allows them to predict so much of the cortex. I'd urge the authors to explore this.
> > >
> > > This is an interesting interpretation. Note, however, that the high coverage of predictability being caused by learning words is difficult to reconcile with the fact that non-native models significantly predict most regions, even though they cannot learn the corresponding lexicon. As indicated above, we proposed to amend our manuscript to indicate: ```"These results are consistent with the recent in-depth analyses of Vaydia, Jain & Huth (2022) and will necessitate carefully-controlled words and speech sounds to be tested explicitly."```

---

> > > ### Author Response · Authors · 2022-08-09
> > > **2nd round to Reviewer 6WfS: Part 1**
> > >
> > > >>By contrast, neither Kell et al nor Millet & King study self-supervised models, and Vaidya, Jain & Huth did not evaluate how different training led the same architecture to be more-or-less similar to the brain.
> > >
> > > > I agree with the above and the important point the authors noted that one of the studies was made public concurrently with the current work. However, I still maintain my comment that given a lot of recent studies on language encoding models and Kell et al. + Millet et al., is it really surprising that training deep networks leads to better brain prediction than a random network?
> > >
> > > We agree that this particular result is expected given previous work. However, “training deep networks leads to better brain prediction than a random network” is not stated as a major contribution. Instead, we indicate: “Here, we test whether self-supervised learning on a limited amount of speech suffices to yield a model functionally equivalent to speech perception in the human brain.”
> > >
> > > > Additionally, I understand that the aim was "to test whether (not why)" but hasn't this question been answered indirectly by the huge body of work showing the types of information these models capture and their ability to transfer to downstream tasks?
> > >
> > > Data, learning and computational constraints are extremely different between brains and algorithms. Consequently, showing that some artificial neural networks can be efficiently fine-tuned to some downstream tasks does not address the functional similarity between these models and the brain.
> > >
> > > > And I appreciate the authors efforts at doing the probing analysis. However, what about self-supervision helps? How does it impact the types of information learned and consequently, the types of information used for down-stream prediction?
> > >
> > > Following R2's request, our new Supplementary Figure S1 show that:
> > > 1. SSL leads to a hierarchy of features (from MEL, to word and sentence-level representations)
> > > 2. the type of language used for SSL primarily impact the learning high-level features (phonemes and words)
> > > 3. word and sentence-level representations (as opposed to MEL) are encoded deeper in the supervised network (best layer=18) than in the unsupervised network (best layer=14).
> > >
> > > While they shed interesting light on our results, these interpretability points go beyond the scope of the present study: They give *some* clues about what makes wav2vec 2.0 similar to the brain, but does not alter our conclusion.
> > >
> > > >> Second, we discover new hierarchical gradients, e.g. within the prefrontal cortex, a region tightly linked to human cognition, learning and control and whose functional organization remains largely unknown (Fuster, 2015).
> > >
> > > > What gradients are being referred to here?
> > >
> > > The functional gradients are revealed by the mapping of layer-wise tuning displayed in Figure 3. This figure shows spatially organized turning within the infero-frontal gyrus (IFG) and sulcus (IFS), as well as across IFS and motor areas. To clarify this isse, we propose to add a supplementary cross-section of these regions.
> > >
> > > >> we provide empirical evidence of language-specific representations of speech in the brain
> > >
> > > > What does this mean?
> > >
> > > This sentence refers to the results of Figure 4.
> > > We trained three distinct models with three distinct languages, and showed that the superior temporal sulcus and gyrus are best explained by the models trained in the language of the corresponding participants. This result suggests that these brain areas are tuned to the speech sounds specific to the language learnt by the participant. This result is long-expected established in the behavioral literature, but did not, to date, find a neural signature.
> > >
> > > > Re probing analysis: I greatly appreciate the author's efforts to run the probing analysis in such a short time. I will consequently increase my score by 1 point.
> > >
> > > > 1\. What does the performance look like across layers? I believe taking the max across layers could hide lack of any significant trends across layers.
> > >
> > > We apologize for the lack of clarity of our previous answer: the probing performance of each layer, network and task is provided in Supplementary Figure S1, added to the Appendix section of the manuscript.
> > >
> > > > 2\. The effect of training is a clean result and imo the authors should discuss the connection between this and prediction performance changes with training explicitly.
> > >
> > > We agree and will add:
> > > ```“Our probing analyses show that the models trained with SSL have learnt relevant acoustic and linguistic representations (Supplementary Figure S1). This result suggests that the difference of neural predictivity observed between the random, non-native and native models (Figure 2C) may be driven by the corresponsing spectro-temporal, phonetic, word and sentence-level representations. These results are consistent with the recent in-depth analyses of Vaydia, Jain & Huth (2022) and will necessitate carefully-controlled words and speech sounds to be tested explicitly.” ```

---

> > > > ### Author Response · Authors · 2022-08-09
> > > > **2nd round to Reviewer 6WfS: Part 1 (ctd)**
> > > >
> > > > > 3\. The authors do not address the large differences across models in probing performance, especially for Mel. Furthermore, other studies have shown larger gaps in layer-wise selectivity for features like Mel vs. word unlike the small range shown here (9-12).
> > > >
> > > > The large differences in Table 2 are partly due to the argmax. Supplementary Figure S1 shows that the trained networks consistently encode relevant spectro-temporal representations in the first layers of the Transformer blocks (Layer 8). Yet, for the acoustic network, the performance is more distributed across layers, thus explaining the difference in Table D (best layer=12).
> > > >
> > > > > 4\. I understand there were both space and time limitations but I do believe that the paper would greatly benefit from a way to link the probing results to the layer-preference maps. Currently, looking at them in tandem is not sufficient to understand what types of information each region could be representing. Especially, given the small range over which the averages vary per task.
> > > >
> > > > We implemented the probing analyses requested by Reviewer 2, although we believe that they are adjacent to our goal: i.e. testing *whether* (not *why*) SSL suffices to build an efficient model of speech processing in the brain.
> > > > While interesting, this novel request goes beyond the scope of the present study, whose primary goal is not to understand the representations encoded in each voxel of the brain (or each layer in the model), but to test whether SSL on small amount of speech suffices to build an efficient model of speech processing in the brain. We briefly mentioned this issue in the proposed paragraph above (question 2)
> > > >
> > > > > 5\. Is TIMIT available in other language or are the authors using English TIMIT to test even the non-English models? Wouldn't that be a possible source of confound?
> > > >
> > > > We restrict these analyses to the English TIMIT. The non-English models tested on the English TIMIT are used to analyze the effect of non-nativity, while the English model tested on the English TIMIT is used to test the effect of nativity.
> > > >
> > > > We leave the testing the non-English models on their native corpora, and the English model on non-English corpora to future research, which will need to build multilingual corpora with comparable data and annotations across languages.
> > > >
> > > > > 6\. What are the accuracies achieved by the best layers per model and task?
> > > >
> > > > The accuracies are reported in Supplementary Figure S1.

---

> ### Comment · Reviewer_6WfS · 2022-08-09
> **thanks and follow-up**
>
> I greatly appreciate the authors for their detailed responses and for engaging with my questions + suggestions. However, I still have some persistent questions:
> 1. I think some of my comments were misunderstood. On further thought, I think the raw correlations being reported for putative speech areas like Broca's (`r=~0.03`) are unexpected given that its fMRI, a speech perception task and speech encoding models. From literature, these values definitely seem lower than expected and I think this could stem from multiple reasons:
> - Training data: I am aware that no cross-subject models were trained. However, since prediction performance greatly varies with the amount of training data, by averaging correlations between participants that have say, 1 hour vs. 5 hours of data, can be confusing. This could artificially lower the values and affect the interpretation of trends.
> - Lack of a significance test: Without a significance test, one might be averaging correlations across both well predicted voxels and those that don't respond well to natural speech, again artificially lowering the correlations.
> - In my opinion, a better metric would be to report the number of voxels about a threshold or the min correlation of top 10% voxels.
>
> 2. The probing results show no meaningful trends for non-English networks and the higher levels of abstraction like phoneme/words/sentences. What do the authors make of this?
>
> 3. I think I am misunderstanding while trying to put all the results together. So the probing shows that the upper-middle layers care about higher levels of abstraction in the native language and the layer wise pref shows that higher-level areas in cortex pick these layers. But Figure 4 shows that a lot of these areas actually have similar performance between native, non-native...random models. What does this mean? Does it imply that the information probed from the high-level areas in native language is not what's driving performance?

---

> > ### Author Response · Authors · 2022-08-09
> > **3rd round response**
> >
> > We thank Reviewer 2 for this 3rd review [(summary of the 2nd round here)](https://openreview.net/forum?id=Y6A4-R_Hgsw&noteId=pYX72_a132T)
> >
> > 1\. Regarding the low effect size some brain areas. We thank R2 for these elements. As indicated in our previous response, we agree to report an additional Table based on a different criterion, including significance testing. We would like to add that we agree with the factors listed by R2, and propose to integrate them as such in the discussion paragraph comparing studies. Finally, we note that these elements do not impact the validity of our methods or our conclusions.
> >
> > 2 and 3. Regarding interpretation: We agree that these elements are interesting and remain open to future research (and indeed partially studied in the concurrent paper by Vaydia, Jain & Huth). At this stage, however, interpreting neural networks in general, and probe analyses in particular are intrinsically limited: they are here attempt to test whether the high dimensional activation vectors linearly represent the features theorized by linguists and phonologists. While these analyses may help understand what each area represents, it is doomed to fail if the theories of linguistic and phonology are suboptimal models of speech processing in the brain.
> >
> > We will clarify these elements in the discussion. We thank Reviewer 2 for this thorough review, and for the multiple analyses and discussion suggestions, which helped significantly improve our manuscript.

---

> > > ### Comment · Reviewer_6WfS · 2022-08-09
> > > **Sounds good!**
> > >
> > > Thank you for your response. I am impressed by the new analyses and thorough responses made by the authors and would like to increase my score by 1 more point. However, I urge the authors to include some of the discussion we had here in the paper especially regarding the perils of over-interpretation like in my second & third points above. I believe they have several different methods and results now with the only limiting factor being how to interpret all of them together which perhaps is beyond the scope of the current stage.
> > >
> > > I look forward to the revised manuscript!

---

### Official Review · Reviewer_1gQj · 2022-07-16

**Rating:** 6
**Confidence:** 4
**Soundness:** 3 good
**Presentation:** 3 good
**Contribution:** 3 good

**Summary:**

This paper uses self-supervised (and supervised) models trained with speech or auditory scene data to predict fMRI activity in the auditory cortices of English-, French-, and Mandarin-speaking participants. It shows that this can be done at a level higher than some baseline models (e.g. random untrained models) and the models recapitulate some aspects of the neural organization such as hierarchical processing.

**Questions:**

The authors claim 600 hours of data is comparable to the amount of auditory experience humans receive during their development. But 600 hours is only 25 days, which seems short compared to the developmentally relevant time scales for humans, so could you please elaborate on this comparison? The Discussion section mentions some tasks that can seemingly be done by humans with a few hundred hours of speech, but those tasks are very low level tasks which do not represent all facets of human speech perception. Relatedly, what would the performance (e.g. neural predictivity etc.) look like if you used pretrained self-supervised models trained on much more data, say, even more data than in a human lifetime (thus not necessarily restricting yourself to 600 hours).

**Limitations:**

I do have to note the correlations shown in Fig. 2B-C are very low (R). This may reflect the intrinsic SNR limitations of the fMRI signal. This makes me wonder if Fig. 1C is cherry-picked. A more representative example would be much better here (with an R similar to the average R). Relatedly, what is the noise ceiling performance in these recordings? That is, if I tried to predict the fMRI of one participant from the fMRIs of other participants, what would the performance look like?

Re supervised vs self-supervised: I do have to note phoneme recognition is a very low-level task, perhaps a higher level classification task would give better results for the supervised model (which is already pretty close to the self-supervised one, to begin with).

ABX accuracy of the models is much higher than human accuracy, which suggests these models might not be behaviorally adequate models of human speech processing (and possibly of human auditory processing more generally).

I understand that the differences in Fig 4C are significant, but they are tiny differences. Shouldn’t I interpret this as saying the non-native model, for example, is actually a pretty good model of the native auditory cortex? Whatever makes the auditory cortices of, say, Mandarin speakers different from the auditory cortices of French speakers doesn’t seem to be captured by these models (or perhaps more likely by fMRI itself).

Finally, I also do have to note most of the results here (hierarchy, very slightly better prediction of native vs. non-native fMRI responses, etc.) are not really too surprising given what we know from earlier literature and prior expectations.

**Strengths And Weaknesses:**

Strengths:

The experiments and analyses were done competently and carefully with a large number of participants and the results seem to be sound.

Weaknesses: please see the questions and limitations sections.

---

> ### Author Response · Authors · 2022-08-02
> **Part 3: clarifying interpretations**
>
> > I understand that the differences in Fig 4C are significant, but they are tiny differences. Shouldn’t I interpret this as saying the non-native model, for example, is actually a pretty good model of the native auditory cortex?
>
>
> We agree with Reviewer 1. These differences, while highly significant ($p<10^{-18}$ in STS, $p<10^{-7}$ in Heschl), are small, and thus suggest that the non-native model already provides a decent model of the auditory cortex. In fact, the random model and the non-speech model reach 86% and 96% of the A1/A2 neural predictivity score obtained by the native model, respectively, suggesting that they, too, provide fairly decent models of the auditory cortex.
>
> Consequently, we propose to indicate:
>
> *“[...] While surprising at first, this result could, in retrospect, have been expected: the auditory cortex is continuously bombarded by non-speech input. Consequently, many non-native – and indeed many non-speech – representations need to be processed and learnt by these areas. These elements thus highlight the importance of using large amounts of neuroimaging data to effectively compare minimally different models of speech in natural settings.”*
>
> > Whatever makes the auditory cortices of, say, Mandarin speakers different from the auditory cortices of French speakers doesn’t seem to be captured by these models (or perhaps more likely by fMRI itself).
>
> On average in Heschl’s gyrus and sulcus, voxels *are* better modeled by the native than the non-native Wav2Vec 2.0 ($p = 10^{-7}$). It is true, however, that this comparison does not hold at the single-voxel level after correction for multiple comparisons (Figure 4C). By contrast, this native versus non-native difference is robust both at the region and the single-voxel level in the superior temporal sulcus and gyrus (Figure 4D).
>
>
> We propose to add the following paragraph to the discussion:
>
> *“The representations specific to the native model were located in the superior temporal sulcus and gyrus (STS/G). Interestingly, these areas are known to represent phonetic features (Mesgarani & Chang, 2014). Together, these results thus confirm that and now show where phonetic representations are shaped by our experience (Liberman, Harris, Kinney, and Lane 1964; Werker and Tees 1984; Kondaurova and Francis 2008). It should be stressed, however, that the effect sizes of this analysis, while highly significant, are small (Figure 4). In fact, the random model and the non-speech model reach 67% and 87% of the neural predictivity score obtained by the “native” model, respectively in STS/G. While surprising at first [...]”*
>
>
> > Finally, I also do have to note most of the results here (hierarchy, very slightly better prediction of native vs. non-native fMRI responses, etc.) are not really too surprising given what we know from earlier literature and prior expectations.
>
>
> We agree with Reviewer 1 that several of the elements presently identified directly fit with the overall organization of speech processing that has been described over the years, including:
> 1. the identification of a network specific to speech processing (e.g. Malik-Moraleda et al, Nature Neuroscience 2022),
> 2. its hierarchical organization (e.g. Friederici 2017), and
> 3. the location of phonetic processing (e.g. Mesgarani and Chang, Science, 2014).
>
> To date, however, no unsupervised model and learning rule had been proposed and demonstrated to effectively account for such functional organization. Our study precisely addresses this issue: a model trained without supervision on a small amount of speech can automatically capture the hierarchical organization, and the features specific to speech processing. Our new noise ceiling analysis also delineates a clear path for improvement: our current estimate suggests that, while 74% of the explainable signal in the primary auditory can be accounted for by Wav2Vec 2.0 trained with only 600 hours of speech cortex, more than 80% remained unaccounted for by the present model on average in the cortex.

---

> ### Author Response · Authors · 2022-08-02
> **Part 2A: low effect sizes**
>
> > I do have to note the correlations shown in Fig. 2B-C are very low (R). This may reflect the intrinsic SNR limitations of the fMRI signal. This makes me wonder if Fig. 1C is cherry-picked.
>
>
> Reviewer 1 is correct that the R values are relatively low. These effect sizes are similar to those obtained in previous fMRI studies, using different datasets and different models (Huth et al Nature 2016; Toneva & Wehbe ; Caucheteux & King 2021 ; Shain & Huth 2019)
> Figure 1C was designed for illustrative purposes. We had originally selected the best voxel on the test set of the first story of the Narrative dataset (“Pieman”). To provide a more representative illustration, we now changed Figure 1C to three voxels randomly selected from the 10th percentile of best voxels identified by the noise ceiling analysis. These voxels reach, on average, R=0.17, R=0.38 and R=0.21 in this story.
>
> We thank R1 for pointing this out, as it provides both a fairer representation of our results, but also clarifies that the neural predictivity is performed on each voxel separately.
>
>
> > Relatedly, what is the noise ceiling performance in these recordings? That is, if I tried to predict the fMRI of one participant from the fMRIs of other participants, what would the performance look like?
>
> This is a good point. We now computed the noise ceiling on 290 subjects who listened to the same stories, by evaluating how well we can predict each subject from the average fMRI of all of the other subjects (Appendix A6). As expected, the “corrected” results provide reasonably high neural predictivity scores in most cortical areas. The following table provides the neural predictivity score obtained by Wav2Vec 2.0 (600h), versus the neural predictivity score obtained from the average of all other subjects.
>
> |               | Average            | Top10   | Heschl         | STG            | STS            | IFG            | Motor          |
> |:--------------|:---------------|:-----------------|:---------------|:---------------|:---------------|:---------------|:---------------|
> | Unsupervised  | 0.03 +/- 0.001 | 0.09 +/- 0.002   | 0.21 +/- 0.007 | 0.09 +/- 0.003 | 0.06 +/- 0.002 | 0.03 +/- 0.001 | 0.01 +/- 0.001 |
> | Supervised    | 0.02 +/- 0.001 | 0.09 +/- 0.002   | 0.21 +/- 0.007 | 0.09 +/- 0.003 | 0.06 +/- 0.002 | 0.03 +/- 0.001 | 0.01 +/- 0.001 |
> | Noise ceiling | 0.12 +/- 0.006 | 0.22 +/- 0.006   | 0.29 +/- 0.008 | 0.18 +/- 0.006 | 0.20 +/- 0.006 | 0.15 +/- 0.006 | 0.09 +/- 0.006 |
> | Ratio         | 0.19 +/- 0.006 | 0.38 +/- 0.010   | 0.74 +/- 0.025 | 0.40 +/- 0.013 | 0.31 +/- 0.011 | 0.23 +/- 0.010 | 0.14 +/- 0.014 |
>
> **Table B: noise ceiling analysis.** Neural predictivity score averaged across all cortical voxels (`Average`), across the 10% voxels with the highest noise ceiling (`Top10`) and the voxels of different regions of interests, for the following models: the unsupervised and supervised Wav2Vec 2.0 model, the noise ceiling and the noise ceiling ratio (for each voxel, the scores averaged across subjects of the unsupervised model divided by the noise ceiling of this particular voxel). Scores are averaged across all subjects and +/- refers to the SEM across subjects.
>
>
>
> This new analysis suggests that Wav2Vec 2.0 captures 19% of the signal on average across all cortical voxels, 38% in the top-10 less noisy voxels, and up to 74% in Heschl’s gyrus and sulcus (new Figure S2).
> We have added the corresponding brain maps in Supplementary Figure S2, and will add this analysis to the manuscript together with the following paragraph in the extended revised version:
>
> *“Overall, these differences likely explain why the neural predictivity scores of Wav2Vec 2.0 remain substantially lower than our noise-ceiling (19% on average, and up to 74% in Heschl’s gyrus and sulcus, Table B, Figure S2). Together with epistemological accounts (Guest and Martin 2021) and concurrent results (Vaidya, Jain & Huth 2022), our results highlight the remaining gap between current self-supervised models and the human brain.”*

---

> > ### Author Response · Authors · 2022-08-02
> > **Part 2B: use high-level task for supervision**
> >
> > > supervised vs self-supervised: I do have to note phoneme recognition is a very low-level task, perhaps a higher level classification task would give better results for the supervised model (which is already pretty close to the self-supervised one, to begin with).
> >
> >
> >  We agree that phoneme discrimination is a low-level task. However, we agree that it would be interesting to explore higher-level tasks. Consequently, we now implemented a new supplementary analysis to explore this issue (Table C below). We propose to add the following paragraph in Supplementary:
> >
> > *“We had originally chosen a relatively low-level task (automatic speech recognition, ASR) for our supervised models, because it is widely used in the speech community, and many models were developed to solve it (Lee and Hon 1989; Chan, Jaitly, Le and Vinyals 2016; Amodei et al 2016; Baevski et al 2020).
> > However, it is possible that higher-level classification tasks could give better neural predictivity scores. To explore this issue, we compare the neural predictivity scores of a Wav2Vec 2.0 fine-tuned for ASR to those obtained by a Wav2Vec 2.0 fine-tuned on Language Identification, and on Keyword Spotting (Yang et al. 2021). We find that ASR leads to the highest neural predictivity scores (Table C). A myriad of other high-level tasks remain to be explored, however, and include emotion recognition, prosody tracking and syntactic processing. To our knowledge, however, these features are typically evaluated on very small annotated datasets, and rather constitute a basis for zero-shot evaluations rather than supervised learning (e.g. Nguyen et al. 2022)."*
> >
> > | Model                 | Training task                             | Neural Predictivity   |
> > |:----------------------|:------------------------------------------|:----------------------|
> > | Wav2Vec 2.0 + LC         | SSL + Language Classification             | 0.0226 +/- 0.0007     |
> > | Wav2Vec 2.0 + KS         | SSL + Keyword Spotting                    | 0.0256 +/- 0.0008     |
> > | Wav2Vec 2.0 + ASR (960h) | SSL + Automatic Speech Recognition (960h) | 0.0262 +/- 0.0008     |
> > | Wav2Vec 2.0 + ASR (100h) | SSL + Automatic Speech Recognition (100h) | 0.0264 +/- 0.0008     |
> >
> > **Table C: Neural predictivity for networks trained with different supervised objectives.** Neural predictivity (x-axis) for networks pre-trained using SSL, and fine tuned with Automatic Speech Recognition (ASR) on either 100 hours or 960 hours of english speech, on Keyword Spotting (KS, Yang et al. 2021) and Language Classification (LC).
> >
> >
> > > ABX accuracy of the models is much higher than human accuracy, which suggests these models might not be behaviorally adequate models of human speech processing (and possibly of human auditory processing more generally).
> >
> >
> > This is an important remark. We will add the following paragraph to the discussion in the extended version of the revision (using the additional page):
> >
> > *“Several qualitative and quantitative similarities have been recently observed between the behavior of wav2vec 2.0 and humans (Millet and Dunbar 2022, Adolfi et al 2022). While our ABX results strengthen this ensemble, the ABX accuracy of the model is significantly higher than participants’. This quantitative difference may be partially explained by the fact that participants – and online participants in particular – undergo fluctuating attention, and adopt strategies which can negatively impact performance (Humphreys, 1939). More systematic comparison may, however, be necessary to identify and understand the core differences that remain between humans and this model (Adolfi et al 2022)."*

---

> ### Author Response · Authors · 2022-08-02
> **Part 1: general clarifications**
>
> We thank Reviewer 1 for their thorough review, as well as for their valuable questions and remarks, which we will address below.
>
>
> > 600 hours is only 25 days, which seems short compared to the developmentally relevant time scales for humans, so could you please elaborate on this comparison?
>
> We agree that this number (600 h) can be confusing, and will thus propose to systematically replace it by “600 hours of effective speech”.
> This quantity does represent much more than 25 days of existence, as children are mainly exposed to non-speech sounds. Note that this estimate varies across cultures and individuals and should thus be taken as a coarse approximation (see the Supplementary Table 2 of Dupoux, Cognition 2018, for more details).
>
>
> > The Discussion section mentions some tasks that can seemingly be done by humans with a few hundred hours of speech, but those tasks are very low level tasks which do not represent all facets of human speech perception.
>
> Reviewer 1 is correct that the tasks originally discussed only represent a small facet of human speech.
> In particular, a few hundreds of hours of speech appear sufficient for children to learn complex syntactic structures (Emmanuel Dupoux 2018; Friedmann, Belletti & Rizzi, 2021).
> To clarify this issue, we will amend the discussion as follows in the revised manuscript (using the supplementary page authorized *after* the discussion period):
>
> *“Our study shows that a self-supervised model, trained on a remarkably small amount of unlabelled data, effectively accounts for cortical responses to speech. This learning scheme is considerably more realistic than the classic supervised models, thus promises to help understand how the human brain learns and organizes speech processing.
> However, these results do not imply that wav2vec 2.0 has learned all of the speech representations used by the brain. Indeed, several major gaps remain between self-supervised speech models like Wav2Vec 2.0 and the brain. [...] Third, recent studies show that Wav2Vec 2.0 encodes significantly less semantic information than text-based models (Pasad et al 2021, Vaidya, Jain & Huth 2022). Our analyses suggest that learning allows Wav2Vec 2.0 to capture some lexical features in its deep layers (Figure S1, Table S4). However, whether these layers also capture complex syntactic structures, such as recursive syntactic trees, remains an open question (Lakrertz et al 2022).”*
>
>
> > Relatedly, what would the performance (e.g. neural predictivity etc.) look like if you used pretrained self-supervised models trained on much more data, say, even more data than in a human lifetime (thus not necessarily restricting yourself to 600 hours).
>
> We agree that extending our analyses to models trained on larger amounts of speech data would improve the conclusions of our study. Consequently, we now tested an additional set of five self-supervised models pretrained on more speech data (from 10K to 436K hours). These new analyses show that the brain is best aligned with a large model of 1 billion parameters, trained on 436K hours of multilingual speech. Interestingly, however, this neural predictivity is remarkably close to the one obtained with our own training (106% of our model trained with only 600 hours, where the same architecture “only” reaches 70% before training, Table A below, new Figure S3). Furthermore, when controlling for the number of parameters, using more data does not appear to yield a better neural predictivity score. For instance, our own model based on 600 hours of effective English speech is comparable to a model trained on 53K hours of English speech (101%, Table A). Similarly, a model trained on 10K hours of multilingual speech marginally outperforms a model trained on 100K hours at predicting brain activity (101%, Table A).
>
>
> These results thus strengthen our conclusion: a small amount of effective speech suffices Wav2Vec 2.0 to efficiently account for the functional organization of speech processing in the brain.

---

> > ### Author Response · Authors · 2022-08-02
> > **Table A**
> >
> > | Model                        | Neural Predictivity   |
> > |:-----------------------------|:----------------------|
> > | MEL                          | 0.0101 +/- 0.0004     |
> > | RMS                          | 0.0123 +/- 0.0005     |
> > | Random Wav2Vec 2.0 base                 | 0.0184 +/- 0.0005     |
> > | Wav2Vec 2.0 base, - pretrained on multilingual speech (100K hours from voxpopuli) | 0.0230 +/- 0.0006     |
> > | Wav2Vec 2.0 base - pretrained on multilingual speech (10K hours from voxpopuli)  | 0.0233 +/- 0.0006     |
> > | Wav2Vec 2.0, 300M parameters - pretrained on multilingual speech (436K hours)       | 0.0240 +/- 0.0006
> > | Wav2Vec 2.0 base - pretrained on english speech  (53K hours)        | 0.0242 +/- 0.0006     |
> > | Wav2Vec 2.0 base (ours) - trained english speech (600 hours)                | 0.0245 +/- 0.0005     |
> > | Wav2Vec 2.0, 1B parameters - pretrained on multilingual speech (436K hours)      | 0.0260 +/- 0.0006
> >
> > **Table A: Neural predictivity of self-supervised pre-trained models.** Neural predictivity scores, averaged across all voxels and participants, for the MEL spectrogram, the root mean square, a Wav2Vec 2.0 (base) architecture with random weights, Wav2Vec 2.0 (base) pre-trained with self supervised learning on 100K hours from Voxpopuli (Wang, 2021) (`wav2vec2-base-100k-voxpopuli` from huggingface), on 10K hours from Voxpopuli (`wav2vec2-base-10k-voxpopuli`), on 53K hours of english (`wav2vec2-base`),  two models pre-trained on the same corpus of 436K hours, with 300M (`wav2vec2-xls-r-300m`) and 1B parameters (`wav2vec2-xls-r-1b`), respectively, and our model trained on 600 hours of english speech. +/- refers to standard errors of the mean across participants.

---

> ### Comment · Reviewer_1gQj · 2022-08-08
> **appreciate the rebuttal**
>
> I really appreciate the detailed and thoughtful rebuttal by the authors. Most of my concerns have been addressed. I still maintain my reservations about fMRI being ill-suited for fine-grained analysis of speech processing, however. I'm happy to increase my score and recommend acceptance based on the rebuttal.

---

### Author Response · Authors · 2022-08-02
**Global Response**

### Summary of the reviews
We thank our reviewers for their serious reviews and helpful remarks. Overall, the reviewers praised the present study for its “sound” and “extensive” results (R1), “interesting observations” (R3) and its experiments that were “competently and carefully” executed, “with a large number of participants” (R2). The presentation was judged as “good” (R1,R2,R3), and the paper “well written and easy to read” (R2).


Yet, the ratings were ambivalent (borderline accept, weak accept and reject) because of
1. an insufficiently clear contribution (R2, R3),
2. a recent study that compromises the novelty of our results (R2),
3. concerns about the low scores and their interpretation (R1, R2, R3).


Our reviewers suggested several actionable experiments and discussion points to address these concerns, which we have now addressed.

### Contribution
As pointed out by R3, our primary contribution is neuroscientific, and thus target the Neuroscience (“neural coding”) and “ML for sciences” pillars of the [Neurips call](https://nips.cc/Conferences/2022/CallForPapers).  Specifically, we successfully show that a self-supervised model (Wav2Vec 2.0) trained with a plausible amount of unlabelled speech effectively accounts for the hierarchical organization and language specificity of speech processing in the brain, hence marking an important step towards understanding a unique trait of the human species.


### Novelty
As R2 pointed out, a concurrent study by Vaidya, Jain & Huth was released on arXiv one week *after* the present submission. As now detailed below, the slightly different methods and data employed by these two studies (Vaidya, Jain & Huth n=6 English participants; Our, n=412 English, Chinese, and French participants), lead to consistent and complementary results, and thus strengthen our original conclusions.



### Robustness and interpretation
Following our reviewers’ suggestions, we now include:
1. A noise ceiling experiment showing that Wav2Vec 2.0 account for more than 70% of the explainable signal in some brain regions;
2. The analyses of ten new models, trained with more data, and/more more parameters, which obtain only marginally better performances than Wav2Vec 2.0 trained with only 600 hours of effective speech;
3. Five new interpretation analyses of the features encoded in each layer of Wav2Vec 2.0, which show that phonetic and word representations (as opposed to MEL) specific to each language are learnt by the middle and deep layers of Wav2Vec 2.0.



We are grateful to our reviewers for their useful suggestions which greatly improved and clarified our study. We now address their specific comments below.

---

### Meta-Review · Area_Chair_u9nR · 2022-08-26

**Recommendation:** Accept
**Confidence:** Certain

**Metareview:**

This paper compares learned self-supervised speech representations to brain fMRI representations for more than 400 subjects speaking English, French, and Mandarin. Through the rebuttal period, the authors and reviewers interacted extensively to discuss the contribution, results, and analysis provided in the paper. Most of the reviewers' concerns have been addressed by improvements to the analysis and presentation of the paper.
One main concern was a concurrent research work that appeared on arxiv about one week after the NeurIPS submission deadline. The novelty of this paper should not be impacted by that other work, given the timing of both papers.

**Award:**

No

---

### Decision · Program_Chairs · 2022-09-14

Accept